# Controlling the Fidelity and Diversity of Deep Generative Models via Pseudo Density

**Shuangqi Li**                                                    *shuangqi.li@epfl.ch*
*School of Computer and Communication Sciences, EPFL*

**Chen Liu**                                                       *chen.liu@cityu.edu.hk*
*Department of Computer Science, City University of Hong Kong*

**Tong Zhang**                                                     *tong.zhang@epfl.ch*
*School of Computer and Communication Sciences, EPFL*

**Hieu Le**                                                        *minh.le@epfl.ch*
*School of Computer and Communication Sciences, EPFL*

**Sabine Süsstrunk**                                               *sabine.susstrunk@epfl.ch*
*School of Computer and Communication Sciences, EPFL*

**Mathieu Salzmann**                                               *mathieu.salzmann@epfl.ch*
*School of Computer and Communication Sciences, EPFL*

**Reviewed on OpenReview:** *https://openreview.net/forum?id=8Vk1Bmg3sY*

## Abstract

We introduce an approach to bias deep generative models, such as GANs and diffusion models, towards generating data with either enhanced fidelity or increased diversity. Our approach involves manipulating the distribution of training and generated data through a novel metric for individual samples, named pseudo density, which is based on the nearest-neighbor information from real samples. Our approach offers three distinct techniques to adjust the fidelity and diversity of deep generative models: 1) Per-sample perturbation, enabling precise adjustments for individual samples towards either more common or more unique characteristics; 2) Importance sampling during model inference to enhance either fidelity or diversity in the generated data; 3) Fine-tuning with importance sampling, which guides the generative model to learn an adjusted distribution, thus controlling fidelity and diversity. Furthermore, our fine-tuning method demonstrates the ability to improve the Frechet Inception Distance (FID) for pre-trained generative models with minimal iterations.

## 1 Introduction

The advent of deep generative models has revolutionized the field of image generation. Key developments marked by Variational Autoencoders (Kingma & Welling, 2013), Generative Adversarial Networks (Goodfellow et al., 2014), and the more emergent diffusion models (Ho et al., 2020; Sohl-Dickstein et al., 2015), have demonstrated an unprecedented capability in producing high-quality images. They have been followed by remarkable results in applications such as classification (Xu et al., 2021; Xu & Le, 2022; Xu et al., 2023a;b), super-resolution (Ledig et al., 2017; Li et al., 2022), face generation (Karras et al., 2019; 2020; 2021), and text-to-image generation (Ramesh et al., 2022; Saharia et al., 2022; Rombach et al., 2022). However, despite these advancements, the critical challenge of balancing fidelity (the realism of the generated images (Xu et al., 2024)) and diversity (the variety in the generated images) remains unaddressed, despite this ability being essential for the practical applicability of these models. Our work focuses on tackling this crucial need for improved control mechanisms in generative models.

To this end, we propose to enhance control over generated images through the probability density of image data. Facing the challenge of directly estimating the density on the complex manifold of image data, we introduce a novel surrogate metric called pseudo density, which leverages the nearest-neighbor information of real samples in the feature space, utilizing an image feature extractor such as a Vision Transformer (Dosovitskiy et al., 2020). By calculating the pseudo density of the generated images, we can estimate their likelihood of appearing in the real distribution. Notably, we observe a correlation between pseudo density and the realism as well as the uniqueness of generated images. Based on the pseudo density of samples, we can modulate the probability of specific samples in both the real data and generated data. Specifically, we introduce three distinct methods to enhance the fidelity and diversity of deep generative models: 1) Per-sample perturbation, allowing precise manipulation of realism and uniqueness in any individual generated image via adversarial perturbation with pseudo density as the objective; 2) Importance sampling during model inference, enabling to enhance the fidelity or diversity in the generated data by accepting with higher or lower probability the generated samples that have higher pseudo density; 3) Fine-tuning with importance sampling, guiding the generative model to learn an adjusted distribution skewed towards data with higher or lower pseudo density, thereby controlling the fidelity and diversity.

Our extensive experiments across diverse datasets and various generative models, including different GANs and diffusion models, demonstrate the effectiveness and generality of our proposed approach in controlling the trade-off between fidelity and diversity. Additionally, our fine-tuning method can be employed to improve the Fréchet Inception Distance (FID) (Heusel et al., 2017) scores by fine-tuning pre-trained models and adjusting their fidelity-diversity trade-off. Aside from the Inception Score (IS) (Salimans et al., 2016) and the Fréchet Inception Distance (FID), as a step towards a more comprehensive evaluation, we employ precision and recall (Kynkäänniemi et al., 2019) as disentangled metrics to separately assess the fidelity and diversity of generative models.

The main contributions of this work can be summarized as follows:

1. We introduce a novel metric, termed pseudo density, for estimating the density of image data. It correlates effectively with the realism and uniqueness of individual images.

2. We propose a per-sample perturbation strategy based on pseudo density that enables adjustment of realism and uniqueness for individual samples.

3. We propose importance sampling based on pseudo density during inference, which controls the fidelity and diversity of generative models at inference time, without re-training or fine-tuning the model.

4. We propose fine-tuning with importance sampling based on pseudo density, which balances the trade-off between the fidelity and diversity of generative models without additional computational overhead during inference.

5. We demonstrate that our fine-tuning method is able to improve the Frechet Inception Distance (FID) of deep generative models by adjusting their precision (fidelity) and recall (diversity) trade-off.

## 2 Background and Related Work

**Generative adversarial networks.** GANs (Goodfellow et al., 2014) are popular generative models to learn the underlying distribution of the given training data and generate new samples. They typically consist of a generator $G(\cdot)$ parameterized by $\theta_g$ that transforms a random vector, called the latent code $z$, into a data sample, and a discriminator $D(\cdot)$ parameterized by $\theta_d$ that aims to distinguish the real training data from the generated samples. Both the generator and discriminator are deep neural networks that can be trained in an end-to-end manner. Therefore, training GANs involves solving a min-max optimization problem by alternatively updating the generator and the discriminator. While the original GANs may suffer from training instability, mode collapse, and poor scalability to high-resolution images, many variants (Arjovsky et al., 2017; Gulrajani et al., 2017; Miyato et al., 2018; Brock et al., 2018; Karras et al., 2018; 2019) have been developed to address these problems and enable impressive high-resolution image generation.

**Diffusion models.** Diffusion models have emerged as a prominent class of generative models, renowned for their ability to create highly detailed and varied images, especially in a text-to-image fashion (Ramesh et al., 2022; Saharia et al., 2022; Rombach et al., 2022). These models, introduced in Sohl-Dickstein et al. (2015), simulate a physical diffusion process in thermodynamics: Starting with an image, noise is incrementally added over several steps until the image is transformed into pure noise. The model then learns to reverse this process, effectively reconstructing the original image from its noised state. Subsequently, the Denoising Diffusion Probabilistic Model (DDPM) (Ho et al., 2020) introduced the $\epsilon$ parameterization and adopted a new architecture based on U-Net (Ronneberger et al., 2015), obtaining significantly improved sample quality. Improved Denoising Diffusion Probabilistic Models (IDDPM) (Nichol & Dhariwal, 2021) refined the noise addition process and optimized the model's architecture to improve both the sampling efficiency and the quality of the generated images. Ablated Diffusion Models (ADM) (Dhariwal & Nichol, 2021) further improved the architecture over IDDPM and demonstrated that diffusion models can achieve image sample quality superior to that of GANs.

**Fidelity and diversity of generative models.** Quantitatively evaluating the performance of generative models is still an open and challenging question. Popular metrics such as the *Inception Score (IS)* (Salimans et al., 2016) and the *Fréchet Inception Distance (FID)* (Heusel et al., 2017) are based on the overall quality (fidelity and diversity) of the generated samples. Kynkäänniemi et al. (2019) proposed the improved precision and recall metrics for generative models to separately evaluate the fidelity and the diversity. The precision is the proportion of generated samples that fall within the manifold estimated by the K nearest neighbors (KNN) of the real data, whereas the recall measures the proportion of real samples that are within the generator-induced manifold. A low precision indicates that many of the generated images do not resemble the training data, suggesting poor fidelity. By contrast, a low recall indicates a lack of diversity in the generated images. An extreme case of low recall is *mode collapse*, i.e., the failure to generate some particular categories of the multi-modal data. Several works have proposed to control the precision (fidelity) and recall (diversity) of generative models, including by truncating the latent distribution (Brock et al., 2018; Karras et al., 2019) and developing latent space samplers (Humayun et al., 2022a;b). Furthermore, Discriminator Rejection Sampling(Azadi et al., 2018) and Discriminator Optimal Transport (Tanaka, 2019) utilize the information retained in the GAN discriminator to improve the fidelity of the generated data during inference. However, these approaches either come with a high computational cost during the inference phase or are limited to specific GAN architectures.

**Realism score and rarity score.** Both the realism score (Kynkäänniemi et al., 2019) and the rarity score (Han et al., 2022), along with our new pseudo density metric, are based on the features extracted from image data using an image feature extractor. To calculate the realism score for an input image, one finds the real image sample that has the highest ratio of its k-NN radius to its distance to the input sample. The rarity score is essentially the smallest possible k-NN radius of a real sample whose k-NN hypersphere contains the input sample. The proposed pseudo density is calculated by a simple neural network that directly fits the estimated densities of real samples in the feature space. Unlike the other two metrics, our metric is specifically tailored for our control approach that manipulates data probability density. Moreover, the realism score is ineffective for samples that are rare in the real distribution, and the rarity score is not computable for outliers that are too far away from other real samples.

## 3 Pseudo Density

Our control approach aims to explicitly manipulate the occurring likelihood of data samples, ideally based on their probability density in the real data distribution. However, estimating the density of high-dimensional data such as images is not straightforward, primarily due to the complexities introduced by the high dimensionality and the fact that image data exist in a low-dimensional manifold. In this section, we introduce an effective metric, termed pseudo density, for estimating the density of image data. Specifically, we first employ a pre-trained image feature extractor to lower the dimensionality of the real image data while capturing the semantics. Then, for each real sample, the average distance $d$ to its nearest neighbors is calculated. Based on the average neighbor distances, we can estimate the volume "occupied" by each sample, and the density is inversely proportional to the volume. Finally, we train a light-weight fully-connected network to fit the estimated density given the extracted feature as input. To compute the pseudo density of any

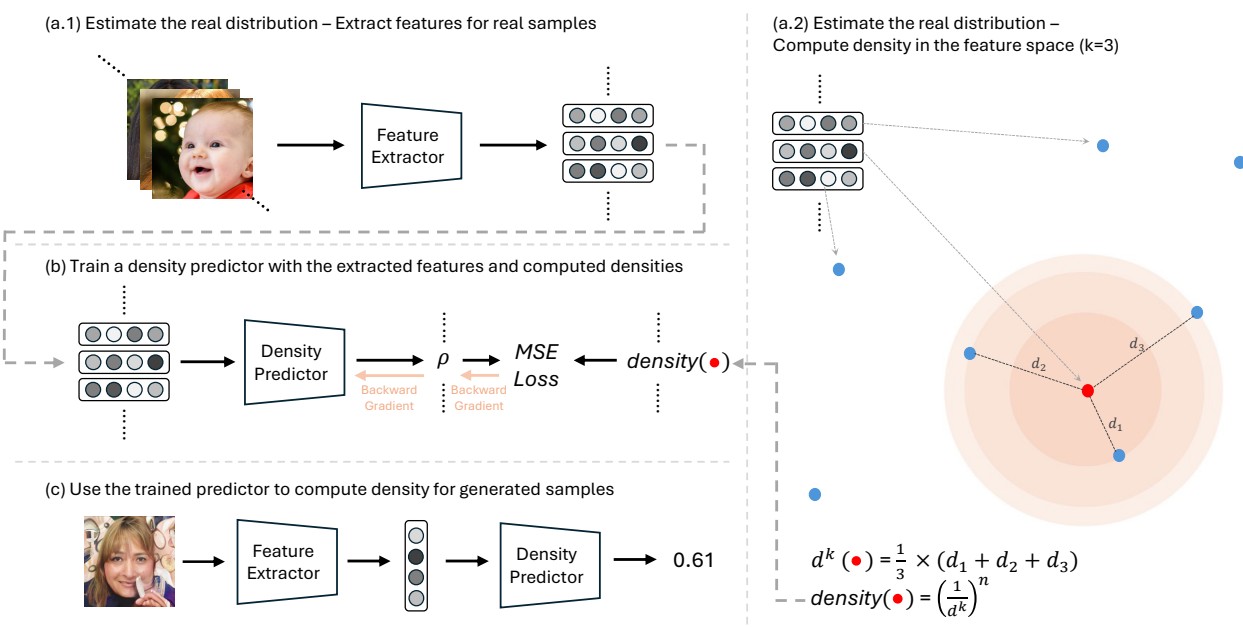

Figure 1: **Overview of the proposed pseudo density metric.** (a) The pseudo-density of each sample is inversely proportional to its average distance to its K-Nearest Neighbors in a feature space. (b) We use these pseudo density to train a density prediction network, which then can be used on any generated samples (c).

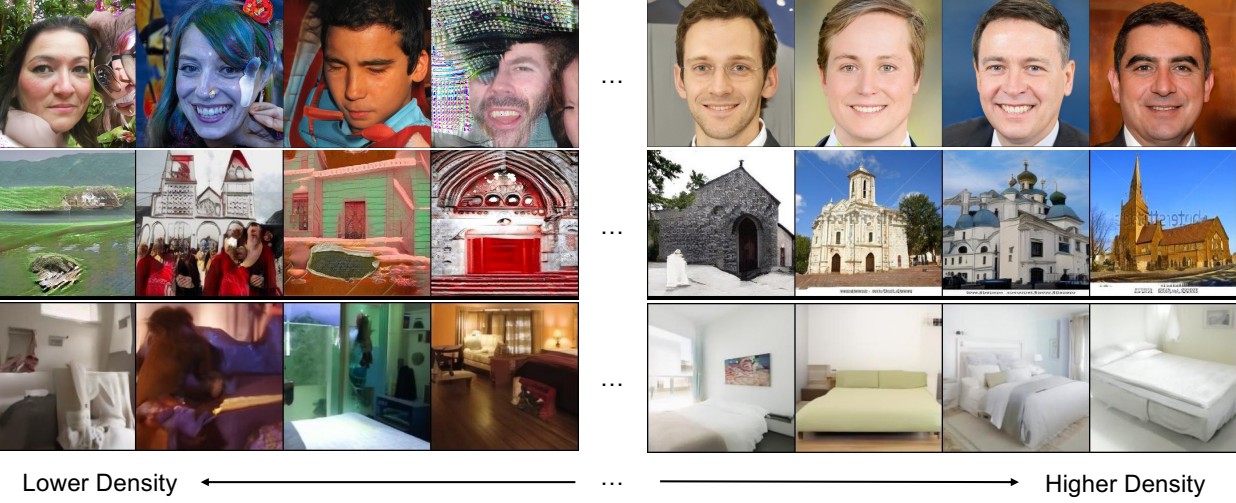

Figure 2: **Images generated by StyleGAN2 (*Top*), ProjectedGAN (*Middle*), and IDDPM (*Bottom*).** The training datasets are FFHQ, LSUN-Church, and LSUN-Bedroom, respectively. For each row, the left four images obtained the lowest pseudo density out of 1000 generations, and the right four obtained the highest.

generated sample, we feed it into the cascade of the feature extractor and the trained network. Note that this pseudo density metric differs from the *Fréchet Inception Distance (FID)* (Heusel et al., 2017): Pseudo density evaluates **individual** samples while the FID evaluates the **overall** performance of the generative model by calculating the distributional distance between the real and generated images. We illustrate the steps of computing pseudo density in Figure 1 and provide more details in the following.

**Estimating the real data distribution.** We estimate the real data distribution in a feature space, using the extracted features of samples from the dataset. First, the features of all real samples in the dataset can

be extracted by a pre-trained feature extractor, such as a Vision Transformer (ViT) (Dosovitskiy et al., 2020) trained on ImageNet (Deng et al., 2009). Then, for each data sample $x_i$, we find its top-$k$ nearest neighbors based on the Euclidean distance in the feature space and compute its average distance to these $k$ samples $d_i^k$. With the assumption that the sample and its neighbors are locally uniformly distributed within a sphere of radius $d$, we can compute the volume a sample "occupies" as $V_i^k = \frac{\pi^{n/2}}{\Gamma(\frac{n}{2}+1)}(d_i^k)^n = C \cdot (d_i^k)^n$, where $n$ is the dimensionality of the feature space, $\Gamma$ is Euler's gamma function, and $C$ is a constant only conditioned on $n$. The density at the data point $x_i$ is hence inversely proportional to the volume, i.e., $\hat{\rho}_i^k \propto \frac{1}{V_i^k}$. Finally, for numerical stability, we compute the normalized density across all data points as $\rho_i = N \cdot \hat{\rho}_i^k \cdot \frac{1}{\sum_i^N \hat{\rho}_i^k}$, where $N$ is the total number of samples. Note that although image data (as well as their extracted features) are in a high-dimensional space, they reside in a manifold of much lower dimensionality. Hence, in practice, we set $n$ to be a small number (e.g., 1 or 2), preventing the volume $V$ from the curse of dimensionality.

**Learning a model that predicts the density.** To efficiently compute the density for any data point $x$ and enable gradient back-propagation, we train a simple regression network $\rho(\cdot)$ to fit the density $\rho_0, \rho_1, ..., \rho_{N-1}$ with the extracted features of real samples as input $F(x_0), F(x_1), ..., F(x_{N-1})$. We observed that a basic fully-connected network consisting of only three hidden layers with fewer than one million parameters is capable of fitting well to the real data while generalizing very effectively. **We term the output of the network as pseudo density.** Compared to directly computing the density of a generated sample by injecting it into all real samples, this approach has lower computational overhead and ensures that the gradient can be back-propagated from the metric to features.

**Compute pseudo density for generated samples.** To calculate the pseudo density of a generated sample $x$, we sequentially feed it to the image feature extractor and the learned regression model $\rho(F(x))$. Intuitively, an image with a high density typically exhibits more common characteristics, whereas an image with a low density is likely to feature more unique attributes. Moreover, in the context of generated images, the lack of training on low-density data often results in reduced realism for low-density images. In Figure 2, we illustrate the effectiveness of pseudo density by showcasing generated images with either high or low pseudo density. For each model under consideration, we generate $1,000$ images, sort them according to their pseudo density, and select the ones with the highest and lowest extremes. The results evidence that the proposed metric aligns well with human perception in terms of realism and uniqueness.

## 4 Density-Based Perturbation

While adversarial perturbation was originally used to attack an input image to fool a classification model, their formulation applies to any other type of input to a neural network as long as an objective function is defined. In this work, we utilize pseudo density as the objective function to adversarially perturb a GAN's latent code or the input noise vector of a diffusion model, both of which we refer to as a latent vector $\boldsymbol{z}$ below.

In the context of image synthesis, such a perturbation optimizes the latent vector within its neighborhood such that it leads to a higher-density or lower-density generated image, hence increasing the realism or uniqueness of the generated image without drastically changing the image content. Let us consider a randomly-sampled latent vector $\boldsymbol{z}$. We explore the neighborhood of $\boldsymbol{z}$ using a perturbation $\delta$ based on the pseudo density. To achieve this, we employ the PGD attack (Madry et al., 2018) strategy to minimize the pseudo density of the features of $G(\boldsymbol{z}+\boldsymbol{\delta})$ under the constraint $\|\boldsymbol{\delta}\|_\infty \leq \epsilon$, where $\epsilon$ is the predefined adversarial budget. The pseudo-code of our per-sample perturbation method is provided in Algorithm 1. In Madry et al. (2018), the perturbation is applied directly to the image and the magnitude of $\epsilon$ should be kept small to ensure that the changes are imperceptible in the image; in our context, however, the perturbation is applied to the latent code and the resulted changes in the image are intended to be noticeable. Additionally, $\epsilon$ determines the magnitude of the pseudo density shift and the extent of the content changes in the generated image. A larger $\epsilon$ yields a large change in pseudo density and also more significant changes in the content and realism/uniqueness of the generated image.

---

**Algorithm 1** Per-Sample Perturbation Based on Pseudo Density

---

**Require:** The image generator $G(\cdot)$, the image feature extractor $F(\cdot)$, the pseudo density function $\rho(\cdot)$, the number of PGD iterations $K$, the step size $\alpha$, the adversarial budget $\epsilon$, and the initial latent vector $\boldsymbol{z}$.

 1: Set an initial perturbation $\boldsymbol{\delta} = \boldsymbol{0}$
 2: **for** $t = 1, ..., K$ **do**
 3:  Calculate the pseudo density of the perturbed sample $G(\boldsymbol{z} + \boldsymbol{\delta})$:
    $f(\boldsymbol{z}, \boldsymbol{\delta}) = \rho(F(G(\boldsymbol{z} + \boldsymbol{\delta})))$
 4:  Update $\boldsymbol{\delta}$:
    $\boldsymbol{\delta} \leftarrow \boldsymbol{\delta} + \alpha\nabla_{\boldsymbol{\delta}} f(\boldsymbol{z}, \boldsymbol{\delta})$ if the goal is to increase realism
    $\boldsymbol{\delta} \leftarrow \boldsymbol{\delta} - \alpha\nabla_{\boldsymbol{\delta}} f(\boldsymbol{z}, \boldsymbol{\delta})$ if the goal is to increase uniqueness
 5:  $\boldsymbol{\delta} \leftarrow clip(\boldsymbol{\delta}, -\epsilon, +\epsilon)$
 6: **end for**
**Ensure:** The perturbed latent vector $\boldsymbol{z} + \boldsymbol{\delta}$.

---

## 5 Density-Based Importance Sampling

With the help of pseudo density, not only can we perturb individual samples, but also manipulate data distributions, assigning adjusted probability to samples with different densities. To achieve this, we proceed as follows. For the given dataset, we estimate its data distribution and compute the pseudo density for all samples, as described in Section 3, and assign two different importance weights for the samples whose densities are above and below a pre-defined density threshold, respectively. Intuitively, by assigning higher weights to above-threshold or below-threshold samples for importance sampling, we can either increase or decrease the proportion of high-density images in the manipulated data distribution.

To edit the generated data distribution in a streaming fashion, one can employ rejection sampling (Azadi et al., 2018). Let the original generator's output distribution given the random latent vector be $p(x)$, and the desired distribution be $q(x)$. Furthermore, let $M$ be the upper bound of the ratio $\frac{q(x)}{p(x)}$, i.e., $\forall x, q(x) \leq Mp(x)$. Rejection sampling proceeds by drawing a sample $x^*$ from $p(x)$ and only accepting $x^*$ with probability $\frac{q(x^*)}{Mp(x^*)}$. In practice, we formulate the desired distribution $q(x)$ by assigning an importance weight $w$ to the samples that have pseudo density higher than a pre-defined threshold $\tau$, and 1 to the other generated samples. In the case where $w > 1$, which means the high-density samples are highlighted, $q(x)$ can be obtained through iterative execution of two steps: 1) Generate a sample from the generator; 2) accept this sample if its pseudo density is higher than the threshold $\tau$, otherwise, reject it with a probability of $1 - \frac{1}{w}$. The case where $w \leq 1$ can be handled similarly by rejection sampling.

### 5.1 Inference with Importance Sampling

A simple yet effective method for controlling the fidelity and diversity of generative models is to employ the above importance sampling strategy to generated images during inference, directly manipulating the generated data distribution. To perform inference in a streaming manner, we employ rejection sampling. Different combinations of density threshold $\tau$ and importance weight $w$ result in different effects on the data distribution, increasing either high-realism samples or high-uniqueness ones. In practice, we set the threshold to be the $\{20, 50, 80\}$ percentile of the pseudo density values of all real samples, and the weight $w$ ranges from 0.01 to 100. Intuitively, adopting an importance weight $w > 1$ leads to more samples with a pseudo density higher than the threshold $\tau$, hence more samples of better realism, while an importance weight $w < 1$ results in more samples with a pseudo density lower than $\tau$, hence more variety of samples and better overall diversity. The pseudo-code of this method is provided in Algorithm 2. We demonstrate our experimental results in Section 6.

### 5.2 Fine-tuning with Importance Sampling

In contrast to our previously introduced method, which manipulates the generated data distribution at inference time, the current method employs importance sampling during the fine-tuning process to adjust

---

**Algorithm 2** Importance Sampling during Inference

---

**Require:** The image generator $G(\cdot)$, the image feature extractor $F(\cdot)$, the pseudo density function $\rho(\cdot)$, the number of images to generate $K$, the density threshold $\tau$, the importance weight $w$ for high-density samples.
1: Set the number of generated images $k \leftarrow 0$.
2: Initialize an empty list to store generated images $I \leftarrow []$
3: **while** $k < K$ **do**
4:     Randomly sample a latent vector $\boldsymbol{z}$ from the prior distribution.
5:     Calculate the pseudo density of the generated sample $G(\boldsymbol{z})$:
            $f(\boldsymbol{z}) = \rho(F(G(\boldsymbol{z})))$
6:     Generate a random number $u \sim \mathcal{U}(0, 1)$
7:     **if** $w > 1$ **then**
8:        **if** $f(\boldsymbol{z}) > \tau$ or $u < \frac{1}{w}$ **then**
9:           Accept the generated sample $G(\boldsymbol{z})$ and push it to the list $I$.
10:          $k \leftarrow k + 1$
11:       **end if**
12:    **else**
13:       **if** $f(\boldsymbol{z}) < \tau$ or $u < w$ **then**
14:          Accept the generated sample $G(\boldsymbol{z})$ and push it to the list $I$.
15:          $k \leftarrow k + 1$
16:       **end if**
17:    **end if**
18: **end while**
**Ensure:** $K$ generated images.

---

the data distribution at training time. As a result, fine-tuning does not introduce any computational overhead during inference. Furthermore, it enables the realism/uniqueness adjustment of any generated images, given their latent vectors. Similarly to the inference-time method, fine-tuning with importance sampling requires the same hyperparameters, the density threshold $\tau$ and the importance weight $w$, to yield different controlling effects. Before fine-tuning, we compute the pseudo density for every sample in the dataset, and then determine the density threshold $\tau$ based on the computed densities of all samples. During fine-tuning, training examples are sampled from the dataset using the importance sampling strategy described before based on the pre-computed pseudo density.

---

**Algorithm 3** Fine-tuning GANs with Importance Sampling

---

**Require:** The batch size $B$, the generator $G(\cdot|\boldsymbol{\theta}_g)$, the discriminator $D(\cdot|\boldsymbol{\theta}_d)$, the number of iterations $T$., the density threshold $\tau$, the importance weight $w$.
1: **for** $t = 1, ..., T$ **do**
2:     Perform importance sampling with $\tau$, $w$ to obtain $B$ real samples $\{\boldsymbol{x}^{(1)}, \ldots, \boldsymbol{x}^{(B)}\}$.
3:     Perform importance sampling with $\tau$, $\frac{1}{w}$ until generating $B$ accepted samples $\{G(\boldsymbol{z}^{(1)}), \ldots, G(\boldsymbol{z}^{(B)})\}$
4:     Update $D$ by minimizing the loss:
            $-\frac{1}{B} \sum_{i=1}^{B} [D(\boldsymbol{x}^{(i)}) - D(G(\boldsymbol{z}^{(i)}))]$.
5:     Perform importance sampling with $\tau$, $\frac{1}{w}$ until generating $B$ accepted samples $\{G(\boldsymbol{z}^{(1)}), \ldots, G(\boldsymbol{z}^{(B)})\}$
6:     Update $G$ by minimizing the loss:
            $\frac{1}{B} \sum_{i=1}^{B} -D(G(\boldsymbol{z}^{(i)}))$.
7: **end for**
**Ensure:** The fine-tuned generator $G$ and discriminator $D$

---

In general, our fine-tuning approach does not modify the training algorithm except for how images are sampled from the data distribution in each iteration. Therefore, it is compatible with any generative models including GANs and diffusion models. Moreover, it can be further enhanced for certain generative models, particularly GANs, for which importance sampling can be applied not only to the real data distribution

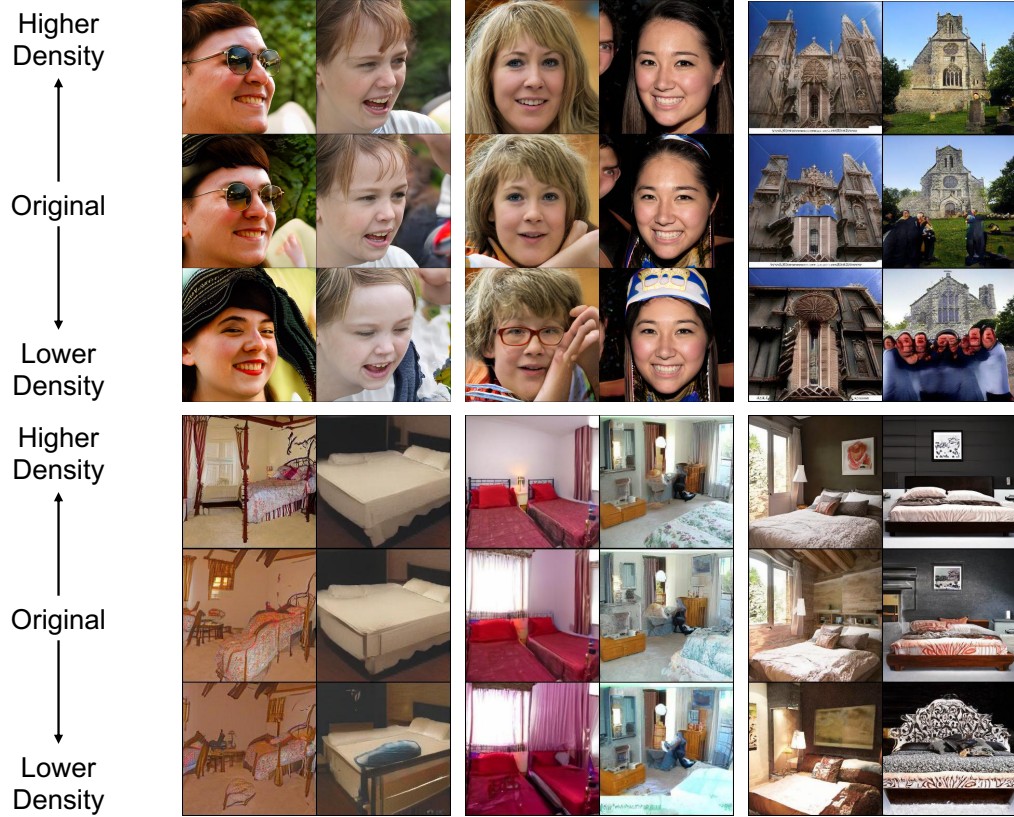

Figure 3: **Per-sample perturbation,** applied to pre-trained models. Given random latent vectors $z$ and their generated images (**Middle** in each panel), we perturb $z$ to achieve higher pseudo density (**Top** in each panel) and lower pseudo density (**Bottom** in each panel). Groups from left to right, and from top to bottom: StyleGAN2 on FFHQ, ProjectedGAN on LSUN-Bedroom; StyleGAN3-T on FFHQ, IDDPM on LSUN-Bedroom; ProjectedGAN on LSUN-Church, ADM on LSUN-Bedroom.

$p_r(x)$, but also to the generated data distribution $p_g(x)$, which serves as input to the discriminator during training. We provide the pseudo-code for the method for GANs in Algorithm 3. Specifically, when the generator and the discriminator reach their equilibrium, we have $\frac{p_r(x)}{p_r(x)+p_g(x)} = \frac{1}{2}$ (Goodfellow et al., 2014), and thus the learned generated data distribution $p_g(x) = p_r(x)$. In the case where we perform importance sampling to the real data distribution $p_r(x)$, we have $p_g(x) = Pert_r(p_r(x))$ upon convergence, where we use $Pert_r(\cdot)$ to represent the perturbation function that maps the original distribution to the sampled one. If we also apply importance sampling to the generated data distribution during training, the equilibrium becomes $Pert_g(p_g(x)) = Pert_r(p_r(x))$. This allows a stronger control over the learned generated data distribution $p_g(x)$ after training. A practical choice is to perform importance sampling on $p_g(x)$ and $p_r(x)$ with reciprocal importance weights $w$ and $\frac{1}{w}$, as in Algorithm 3.

Note that the proposed fine-tuning strategy can also be applied to training from scratch. However, we observed that applying fine-tuning yields comparable effectiveness but with significantly less computational overhead compared to training a model from scratch.

Additionally, our fine-tuning strategy is compatible with inference-time techniques, including truncation and the proposed importance sampling strategy detailed in Section 5. When inference-time methods are applied in conjunction with our fine-tuning strategy, a stronger enhancement over fidelity or diversity can be achieved. Empirical evidence supporting this combined effect is presented in Appendix A.4.

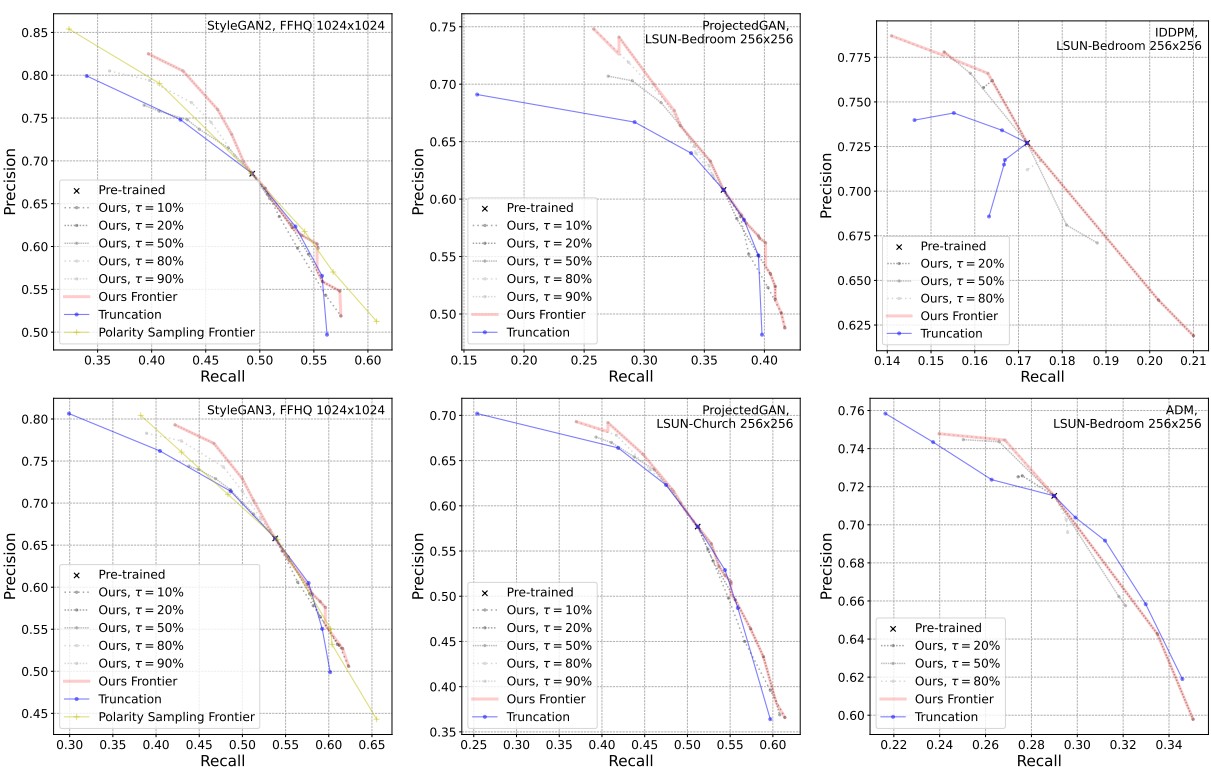

Figure 4: **Precision-recall trade-off.** For all methods, the precision and recall metrics compete with each other. For polarity sampling, we used the values reported in Humayun et al. (2022a) due to replication difficulty. For our method, we adopted different density thresholds $\tau$ for each dashed line with varying importance weights $w$ ranging from 0.01 to 100. For better visualization, we only report part of the results for polarity sampling and truncation. We visualize the Pareto frontier for polarity sampling and ours, illustrating the optimal trade-offs achieved across various hyperparameter configurations.

## 6 Experiments

In this section, we present the results of our methods on various generative models and datasets. We select Improved Denoising Diffusion Probabilistic Model (IDDPM) (Nichol & Dhariwal, 2021) in addition to Ablated Diffusion Model (ADM) (Dhariwal & Nichol, 2021), StyleGAN2 (Karras et al., 2020), StyleGAN3 (Karras et al., 2021), and ProjectedGAN (Sauer et al., 2021) to demonstrate the generality of our methods on different types of generative models. The datasets consist of LSUN-Bedroom (Yu et al., 2015), LSUN-Church (Yu et al., 2015), and FFHQ (Karras et al., 2019). The images from LSUN-Bedroom and LSUN-Church are resized to $256 \times 256$, while the FFHQ images are of resolution $1024 \times 1024$. For all experiments, we use the pre-trained checkpoints provided by the authors and use the same hyper-parameters as in the original papers for fine-tuning. More details regarding the datasets are deferred to the supplementary material, as well as more examples of generated images.

In Figure 3, we showcase examples of images generated by the pre-trained models to illustrate the effects of our proposed per-sample perturbation strategy, which achieves large density change while preserving content of the images. By applying perturbation that increases density, the generated images exhibit greater realism, characterized by simplified backgrounds, fewer objects, etc. By contrast, perturbation to lower density leads to images with higher uniqueness, featuring more complex backgrounds, the presence of rare features in the datasets, etc. In Appendix A.2, we consider perturbation that increases the output of a GAN's discriminator and demonstrate its inferior performance compared to pseudo density. Additionally, we compare our approach with Discriminator Rejection Sampling (Azadi et al., 2018) and Discriminator

Table 1: **Improved precision/recall/FID obtained by fine-tuning with importance sampling, with different importance weights.** *P* stands for precision and *R* stands for recall. **Bold** denotes the best value in each column.

| Model | P↑ | R↑ | FID↓ | Model | P↑ | R↑ | FID↓ |
|---|---|---|---|---|---|---|---|
| **LSUN-Bedroom 256 × 256** | | | | **FFHQ 1024 × 1024** | | | |
| Improved DDPM | 0.727 | 0.172 | 10.13 | StyleGAN2 | 0.685 | 0.493 | 2.79 |
| + fine-tune ($w = 33.0$) | **0.749** | 0.168 | 9.54 | + fine-tune ($w = 33.0$) | **0.811** | 0.387 | 36.24 |
| + fine-tune ($w = 0.03$) | 0.648 | **0.231** | 13.48 | + fine-tune ($w = 0.03$) | 0.558 | **0.564** | 10.58 |
| + fine-tune ($w = 2.0$) | 0.737 | 0.175 | **9.53** | + fine-tune ($w = 0.5$) | 0.675 | 0.500 | **2.60** |
| ADM | 0.715 | 0.290 | 6.35 | StyleGAN3-T | 0.658 | 0.538 | 2.81 |
| + fine-tune ($w = 33.0$) | **0.729** | 0.284 | 7.70 | + fine-tune ($w = 33.0$) | **0.824** | 0.382 | 60.89 |
| + fine-tune ($w = 0.03$) | 0.627 | **0.343** | 8.15 | + fine-tune ($w = 0.03$) | 0.482 | **0.624** | 22.34 |
| + fine-tune ($w = 2.0$) | 0.697 | 0.302 | **5.83** | + fine-tune ($w = 1.5$) | 0.657 | 0.535 | **2.73** |
| ProjectedGAN | 0.608 | 0.366 | 2.31 | **LSUN-Church 256 × 256** | | | |
| + fine-tune ($w = 33.0$) | **0.694** | 0.290 | 3.00 | ProjectedGAN | 0.577 | 0.512 | 1.62 |
| + fine-tune ($w = 0.03$) | 0.524 | **0.419** | 7.65 | + fine-tune ($w = 33.0$) | **0.658** | 0.418 | 3.95 |
| + fine-tune ($w = 1.05$) | 0.617 | 0.359 | **2.22** | + fine-tune ($w = 0.03$) | 0.494 | **0.540** | 7.47 |
| | | | | + fine-tune ($w = 2.0$) | 0.586 | 0.507 | **1.58** |

Optimal Transport (Tanaka, 2019), both of which rely on the trained discriminator's output in GANs as an indicator for image realism.

In Figure 4, we demonstrate the trade-off between precision (fidelity) and recall (diversity) achieved by density-based importance sampling during inference. Our approach achieves a better overall precision-recall trade-off than the other two inference-time methods, polarity sampling (Humayun et al., 2022a) and truncation (Karras et al., 2019), especially in the improved-precision area. It's important to note that polarity sampling may face compatibility issues with diffusion models due to the computational complexity of the denoising process. The truncation method is adaptable to the intermediate $\mathcal{W}+$ space for StyleGAN (Karras et al., 2019; 2020; 2021) models, but is generally limited to the input $\mathcal{Z}$ space for others, which could result in suboptimal performance. For instance, for Improved DDPM, the truncation method struggles around the pre-trained trade-off point. Unlike these methods, neither the computational complexity nor particular model architectures restricts the applicability of our method since our approach directly manipulates the output data.

In Table 1, we demonstrate that our fine-tuning approach can improve the precision, recall, or FID across various datasets and generative models of different types, depending on the choices of hyperparameters, particularly the importance weight $w$. With larger $w > 1$, the fine-tuning procedure encourages the models to generate higher-density data, and vice versa with $w < 1$. By fine-tuning with $w$ close to 1, the model can end up at a slightly altered trade-off point that leads to improved FID scores. Note that the density threshold $\tau$ that we use may vary for the same model. We refer the reader to the appendix for more details about the hyperparameters and fine-tuning configurations. In Figure 5, we further present examples of images generated by the pre-trained models compared with images produced by their fine-tuned versions to visually demonstrate the effects of our fine-tuning method.

## 7 Conclusion

In this study, we have introduced a simple and effective approach for estimating the density of image data, enabling us to devise inference-time methods and a fine-tuning strategy for biasing deep generative models into outputting data with either higher fidelity or higher diversity. We have shown that our method can greatly improve the fidelity or diversity in the generated data without significantly altering their primary subject and structure, making it of great interest to applications such as image editing. We have also

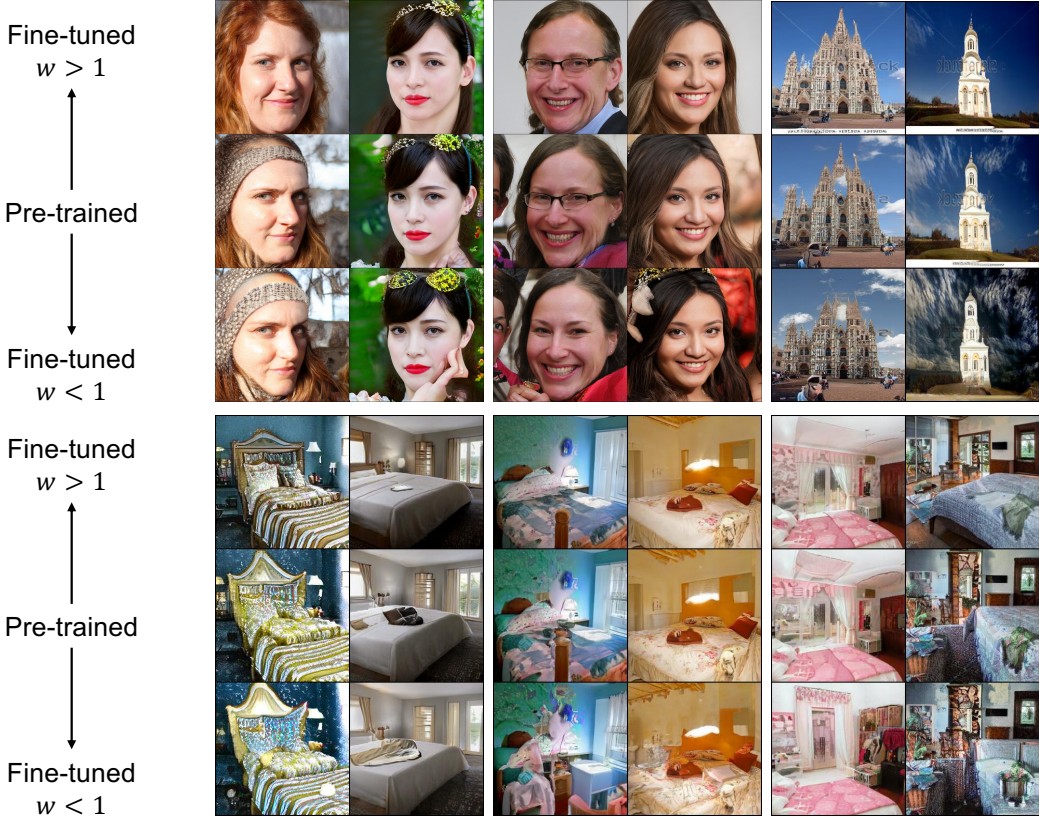

Figure 5: **Images generated by pre-trained and fine-tuned models.** For each column in a panel, the same latent vector is used by the pre-trained model and the fine-tuned versions for image generation. Fine-tuning with $w > 1$ biases the generative model to output more high-density data. Groups from left to right, and from top to bottom: StyleGAN2 on FFHQ, ProjectedGAN on LSUN-Bedroom; StyleGAN3-T on FFHQ, IDDPM on LSUN-Bedroom; ProjectedGAN on LSUN-Church, ADM on LSUN-Bedroom.

demonstrated that adjusting the balance between precision (fidelity) and recall (diversity) through fine-tuning can improve the Frechet Inception Distance (FID) for models pre-trained in a standard manner. This underscores the importance of considering both fidelity and diversity in the evaluation of generative models instead of relying solely on FID as a performance metric. Future research may aim to evaluate and improve various density-based sampling strategies for optimized their efficacy. Another potential research interest lies in adapting the proposed control approach to conditional generation tasks such as text-to-image synthesis.

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

# A  Appendix

## A.1  Training Details, Evaluation Details, and Hyper-parameters

**Datasets** The FFHQ (Flickr-Faces-HQ) dataset (Karras et al., 2019) is a high-quality collection of human face images consisting of 70k images with the resolution of $1024 \times 1024$. The LSUN-Church and LSUN-Bedroom are subsets of the LSUN (Large-scale Scene Understanding) dataset (Yu et al., 2015). For LSUN-Church, we use all 126k images in the dataset and resize them to $256 \times 256$ resolution. For LSUN-Bedroom, we sample the first 200k images from the dataset and apply center crop, resizing them to $256 \times 256$ resolution as well. We followed the same pre-processing procedures as in previous works.

**Training details** We follow the same training hyper-parameters published in the original papers, except the number of GPUs, while keeping the same batch size per GPU. For GANs models in our experiments, we used 2 GPUs, and a single GPU for diffusion models. For StyleGAN2/3 models, we fine-tuned for, measured in thousands of real images fed to the discriminator, 80 thousand images. For Projected GAN, we fine-tuned for 40 thousand images. To improve FIDs, we only fine-tuned all GANs models for 12 thousand images. As for diffusion models in our experiments, we fine-tuned them for $128 \times 20000 = 2560k$ images.

Regarding the computation of pseudo density, we use $n = 1$ for all datasets and $K = 10$ for all except LSUN-Bedroom where we set $K = 20$.

**Evaluation details** To evaluate the FID, precision, and recall for a generative model, we use the model to generate $50,000$ images every time and then calculate the metrics against the training dataset. For diffusion models, however, we only generate $10,0000$ images every time for computational efficiency. In addition, we use the uniform stride from DDIM (Song et al., 2020) and each sampling process takes 50 steps.

**Per-sample perturbation hyper-parameters** We employed PGD attack (Madry et al., 2018) whose hyper-parameters involve the number of steps $K$, the step size $\alpha$, and the adversarial budget $\epsilon$. For all GANs models in our experiments, we adopted $K = 10, \alpha = 0.025$, and $\epsilon = 0.1$. For diffusion models, we adopted $K = 5, \alpha = 0.0025$, and $\epsilon = 0.0125$.

**Density-based importance sampling hyper-parameters** The two relevant hyper-parameters are the density threshold $\tau$ and the importance weight $w$ of above-threshold samples. Their optimal values may vary across different datasets and models. We conducted a sweep for $\tau$ with values from $\{20, 50, 80\}$ percentiles of the estimated densities of real images. We also performed sweeps for $w$ with values from $\{0.01, 0.03, 0.1, 10, 33.0, 100\}$. We show the optimal values found in Table 2. When aiming to improve FIDs, we kept a density threshold of 50 percentile and performed importance sampling only on the real data. We conducted a sweep for $w$ from $\{0.5, 0.6, 0.7, 0.8, 0.9, 0.95, 1.05, 1.1, 1.3, 1.5, 1.7, 2.0\}$ for GANs models and only $\{0.5, 2.0\}$ for diffusion models.

Table 2: **Optimal hyper-parameters for importance sampling.** Optimal hyper-parameters vary with different goals, which are improving precision (Precision$\nearrow$), improving recall (Recall$\nearrow$), and improving FID (FID$\searrow$). In the first two scenarios, the selection was not based on maximizing the precision/recall gain but on the (subjective) optimal trade-off.

| | Precision$\nearrow$ | | Recall$\nearrow$ | | FID$\searrow$ | |
|---|---|---|---|---|---|---|
| Model, Dataset | $\tau$ | $w$ | $\tau$ | $w$ | $\tau$ | $w$ |
| StyleGAN2, FFHQ | 80% | 33.0 | 20% | 0.03 | 50% | 0.5 |
| StyleGAN3-T, FFHQ | 80% | 33.0 | 20% | 0.03 | 50% | 1.5 |
| ProjectedGAN, LSUN-Church | 50% | 33.0 | 50% | 0.03 | 50% | 2.0 |
| ProjectedGAN, LSUN-Bedroom | 50% | 33.0 | 50% | 0.03 | 50% | 1.05 |
| IDDPM, LSUN-Bedroom | 80% | 33.0 | 20% | 0.03 | 50% | 2.0 |
| ADM, LSUN-Bedroom | 80% | 33.0 | 20% | 0.03 | 50% | 2.0 |

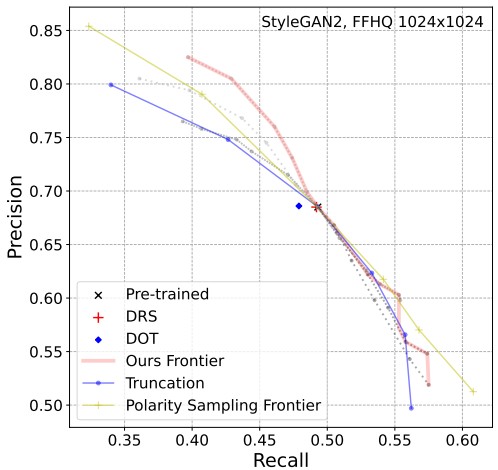

Figure 6: **Precision-recall trade-off** for StyleGAN2 on FFHQ.

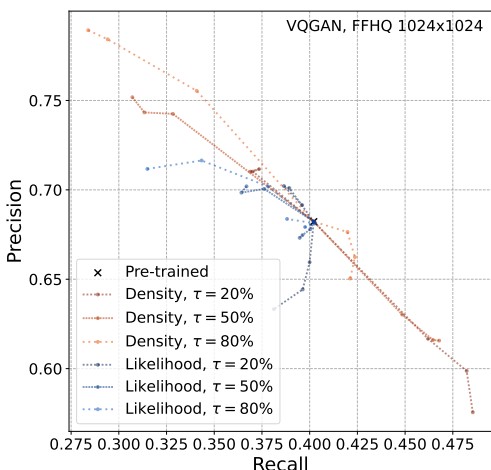

Figure 7: **Precision-recall trade-off** for VQGAN on FFHQ.

## A.2 Discriminator Output versus Pseudo Density

Many works, including LOGAN (Wu et al., 2019), Discriminator Rejection Sampling (Azadi et al., 2018), and Discriminator Optimal Transport (Tanaka, 2019), have explored the output of a trained discriminator as a metric for sample realism. Ideally, discriminator output may indicate sample realism in the context of Generative Adversarial Networks since the discriminator learns to tell if a sample is real or generated. DRS (Azadi et al., 2018) and DOT (Tanaka, 2019) demonstrated that discriminator output aligns well with realism in low-dimensional datasets and their methods improve FID for SNGAN (Miyato et al., 2018) and SAGAN (Zhang et al., 2019). However, our experiments suggested that it works badly for higher-resolution image data and larger modern GAN architectures such as StyleGAN (Karras et al., 2019; 2020). In Figure 8, we showcased the results of per-sample perturbation based on pseudo density and discriminator output using StyleGAN2 on FFHQ dataset. Additionally, we compared to random perturbation, which replaces the gradient-descent direction with a random direction while keeping the same magnitude in each step of the PGD attack.

We implemented DRS and DOT for StyleGAN2 and followed the original hyperparameters reported in the papers. As shown in Figure 6, where we plot the precision and recall for DRS and DOT alongside our approach and others, both DRS and DOT achieved performances similar to using the pre-trained model naively. Additionally, DRS and DOT obtained FID scores of 2.95 and 3.44, respectively, which are worse than that of the pretrained. These results indicate that discriminator-based methods like DRS and DOT do not achieve ideal precision-recall trade-off for high-resolution image data and larger, modern GAN models.

## A.3 Extra Computational Overhead

The proposed rejection sampling strategies, as described in Section 5, Algorithm 2 and Algorithm 3, result in extra computational cost since only a proportion of the generated samples are accepted. Given the importance weight $w < 1$ and let the density threshold be $p\%$ of estimated densities of the data, assuming the generated data perfectly fit the real data, a generated sample is accepted with probability $0.01p + (1 - 0.01p) \times w$ on average, hence the computational cost is approximately $\frac{1}{0.01p + (1 - 0.01p) \times w}$ times as before. The case where $w > 1$ can be handled similarly.

In our experiments, generating samples using StyleGAN2 with importance sampling ($p = 50, w = 0.01$) slowed down from the original 13 sec/kimgs to 26 sec/kimgs while generating with $p = 20, w = 0.01$ slowed down to 60 sec/kimgs. In contrast, generating samples with DRS (Azadi et al., 2018) and DOT (Tanaka, 2019) slowed down to 23 sec/kimgs and 290 sec/kimgs respectively. We were unable to obtain statistics for polarity sampling (Humayun et al., 2022a) due to difficulties in replicating the method, but the authors

reported that preprocessing for StyleGAN2 took a daunting 14 days. In comparison, our method and DOT took several minutes, while DRS took less than an hour.

On the other hand, our fine-tuning method maintains the original inference efficiency but compromises training efficiency. Fine-tuning StyleGAN2 using importance sampling with $p = 80, w = 33.0$ lowered the speed from 56 sec/kimgs to 74 sec/kimgs. However, fine-tuning diffusion models did not exhibit a significant slowdown in training speed since it involves importance sampling only on the real data, avoiding the time-consuming rejection sampling. Notably, our fine-tuning process typically requires less than one hour to complete due to the small number of iterations needed.

### A.4 Combination of Fine-tuning and Inference-Time Methods

Not only the pre-trained models but also the models fine-tuned by our methods can be readily combined with any inference-time methods such as our proposed importance sampling strategy and truncation (Karras et al., 2019) to further boost the precision/recall improvement. In Table 3, we demonstrate that applying them to fine-tuned models can achieve even higher precision/recall than either fine-tuning or solely applying inference-time methods.

Table 3: **Applying truncation or our importance sampling strategy during inference further improves the precision/recall.** $P$ stands for precision and $R$ stands for recall. **Bold** denotes the best value in each column.

| Model | P↑ | R↑ |
|---|---|---|
| StyleGAN2 | 0.685 | 0.493 |
|   + Fine-tune ($\tau = 80\%, w = 33.0$) | 0.811 | 0.387 |
|   + Truncation ($\Phi = 0.7$) | 0.854 | 0.233 |
|   + Sampling ($\tau = 90\%, w = 100$) | 0.825 | 0.397 |
|   + Fine-tune ($\tau = 80\%, w = 33.0$) + Truncation ($\Phi = 0.7$) | **0.938** | 0.163 |
|   + Fine-tune ($\tau = 80\%, w = 33.0$) + Sampling ($\tau = 90\%, w = 100$) | 0.872 | 0.198 |
|   + Fine-tune ($\tau = 20\%, w = 0.03$) | 0.558 | 0.564 |
|   + Truncation ($\Phi = 1.3$) | 0.491 | 0.571 |
|   + Sampling ($\tau = 20\%, w = 0.01$) | 0.548 | 0.574 |
|   + Fine-tune ($\tau = 20\%, w = 0.03$) + Truncation ($\Phi = 1.3$) | 0.386 | **0.617** |
|   + Fine-tune ($\tau = 20\%, w = 0.03$) + Sampling ($\tau = 20\%, w = 0.01$) | 0.473 | 0.597 |

### A.5 Autoregressive Generative Models

Autoregressive generative models, such as PixelCNN (Van den Oord et al., 2016) and VQGAN (Esser et al., 2021), have the unique capability of generating images while providing their exact likelihood in the learned distribution at the same time. This suggests the potential use of sample likelihood as an indicator of image realism, potentially preventing the need for pseudo density computation in the proposed importance sampling strategies. To investigate this possibility, we compared pseudo-density-based and log-likelihood-based importance sampling using the autoregressive VQGAN (Esser et al., 2021). As shown in Figure 7, log-likelihood-based sampling yields an inferior precision-recall tradeoff compared to pseudo-density-based sampling and fails to improve the recall metric even when precision drops below that of the pretrained model. These results indicate that while images with higher likelihood tend to exhibit better realism, those with lower likelihood do not necessarily demonstrate greater uniqueness. We conclude that the likelihood metric provided by autoregressive models is less effective than pseudo density for controlling fidelity and diversity in generated samples.

### A.6 Additional Qualitative Result

In Figures 9, 10, 11 12, 13, 14, we present images generated by per-sample perturbation, compared to the original generated images. In Figures 15, 16, 17, 18, 19, 20, we present images generated by pre-trained models and fine-tuned ones.

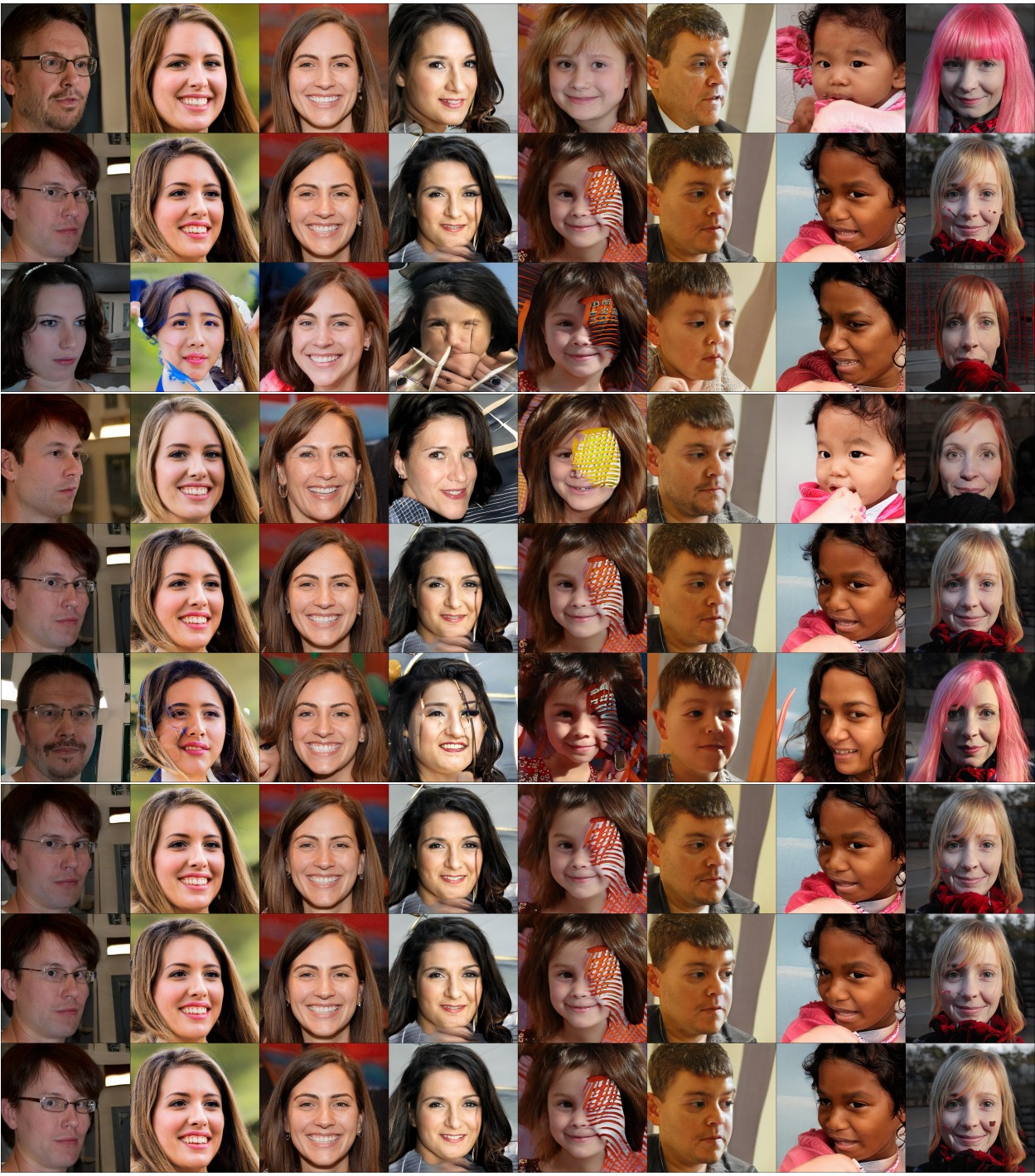

Figure 8: **Images generated by StyleGAN2 pre-trained on FFHQ with per-sample perturbation.** (**Top panel**): Pseudo Density as the objective for PGD. (**Middle panel**): Discriminator output as the objective for PGD. (**Bottom panel**): Random perturbation. In each panel, the middle row displays images generated by the pre-trained model, and the top row corresponds to images perturbed for better realism while the images in the bottom row are perturbed for better diversity.

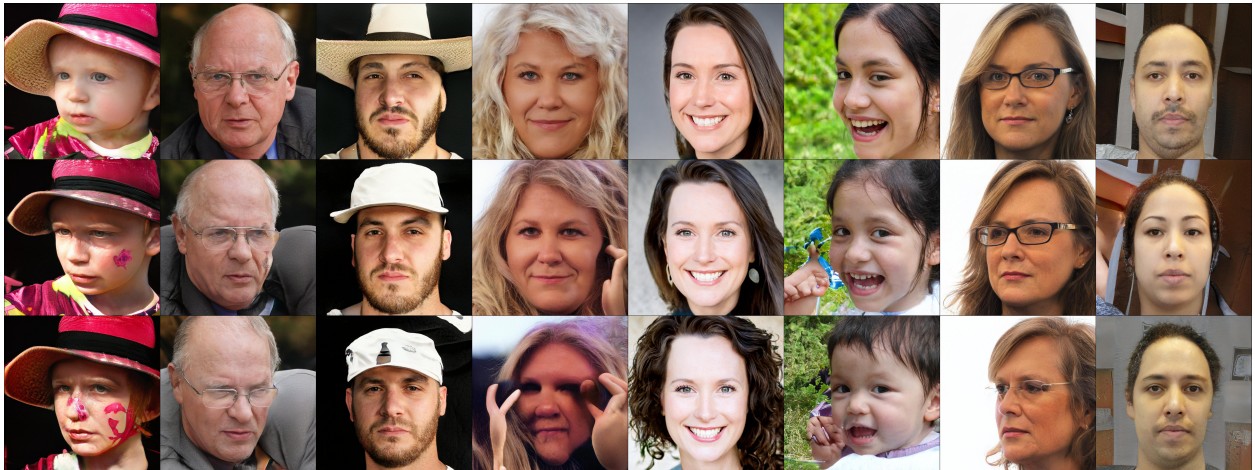

Figure 9: **Images generated by StyleGAN2 pre-trained on FFHQ with per-sample perturbation.** (**Middle**): Images generated by the pre-trained model. (**Top**): Generated by the same model with perturbed latent vectors for better realism. (**Bottom**): Generated by the same model with perturbed latent vectors for better diversity.

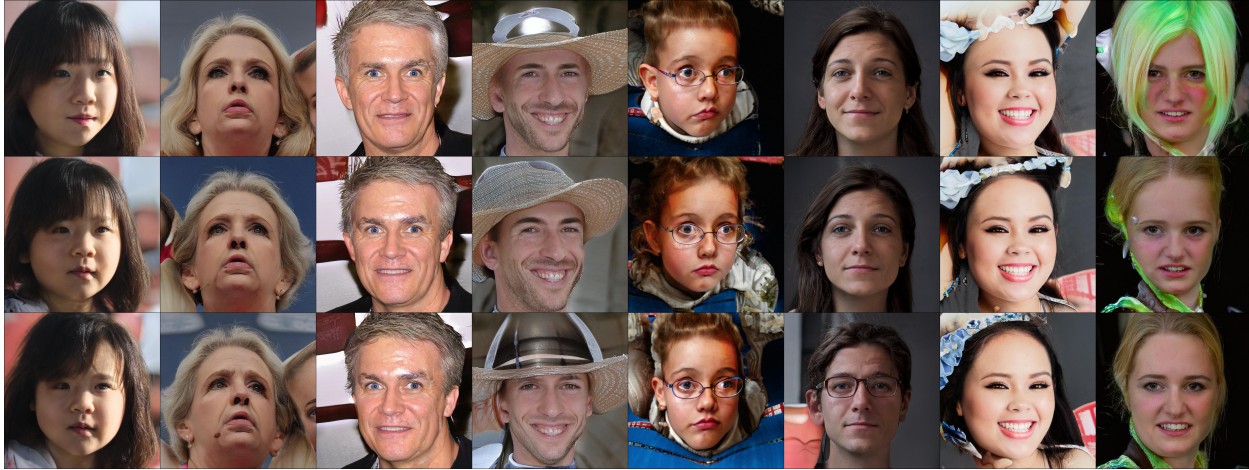

Figure 10: **Images generated by StyleGAN3-T pre-trained on FFHQ with per-sample perturbation.** (**Middle**): Images generated by the pre-trained model. (**Top**): Generated by the same model with perturbed latent vectors for better realism. (**Bottom**): Generated by the same model with perturbed latent vectors for better diversity.

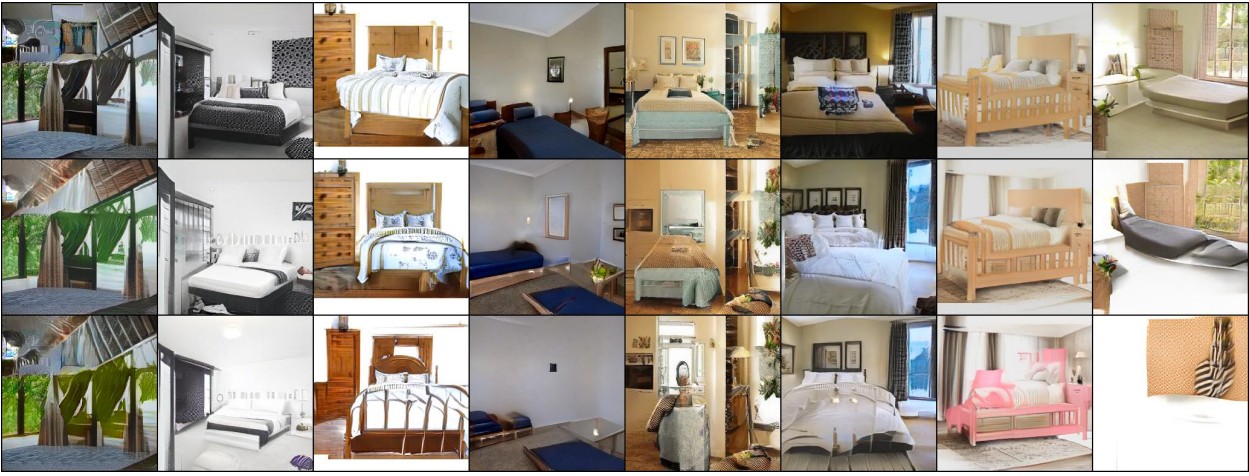

Figure 11: **Images generated by ProjectedGAN pre-trained on LSUN-Bedroom with per-sample perturbation.** (**Middle**): Images generated by the pre-trained model. (**Top**): Generated by the same model with perturbed latent vectors for better realism. (**Bottom**): Generated by the same model with perturbed latent vectors for better diversity.

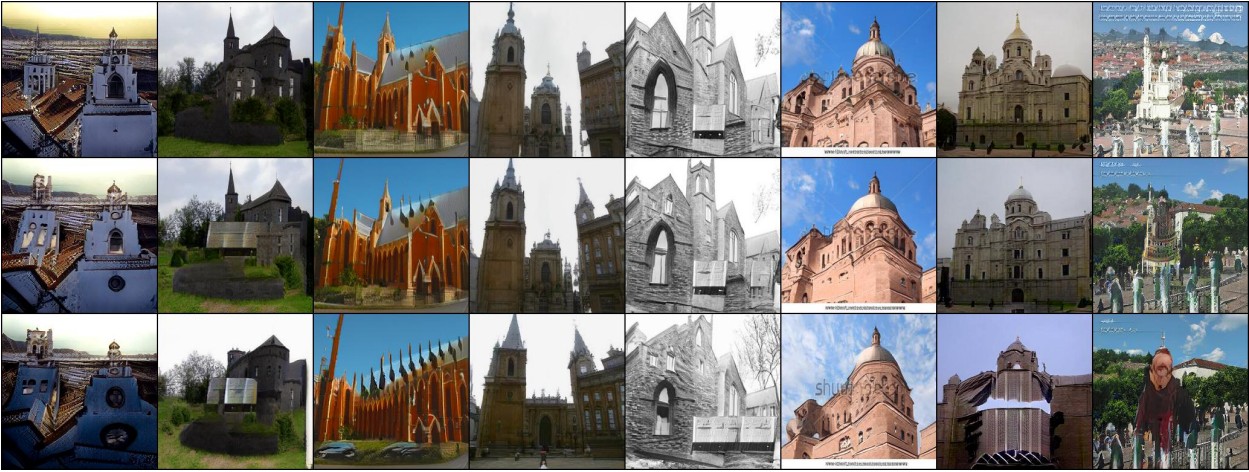

Figure 12: **Images generated by ProjectedGAN pre-trained on LSUN-Church with per-sample perturbation.** (**Middle**): Images generated by the pre-trained model. (**Top**): Generated by the same model with perturbed latent vectors for better realism. (**Bottom**): Generated by the same model with perturbed latent vectors for better diversity.

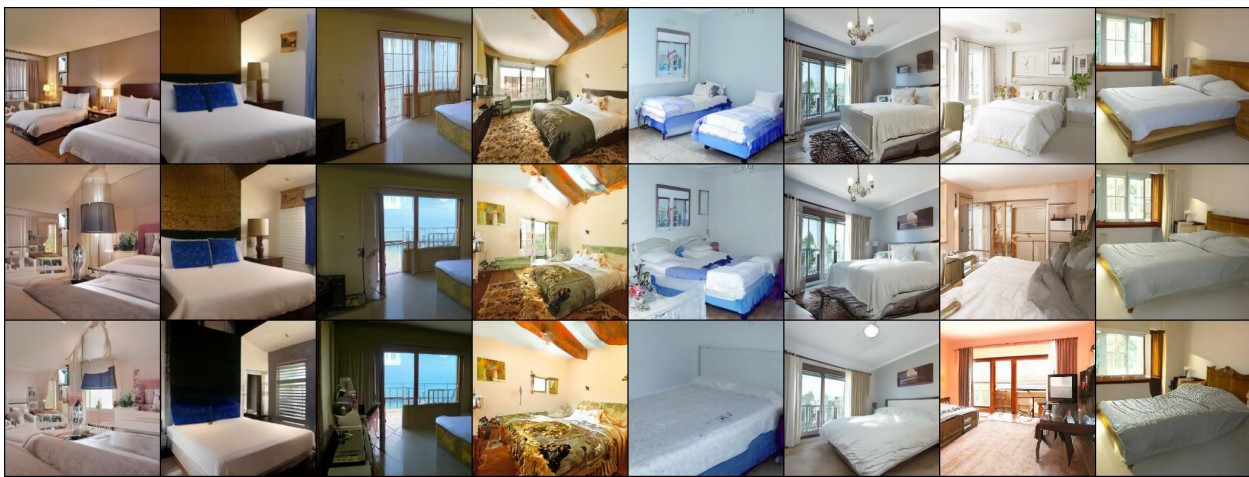

Figure 13: **Images generated by IDDPM pre-trained on LSUN-Bedroom with per-sample perturbation.** (**Middle**): Images generated by the pre-trained model. (**Top**): Generated by the same model with perturbed latent vectors for better realism. (**Bottom**): Generated by the same model with perturbed latent vectors for better diversity.

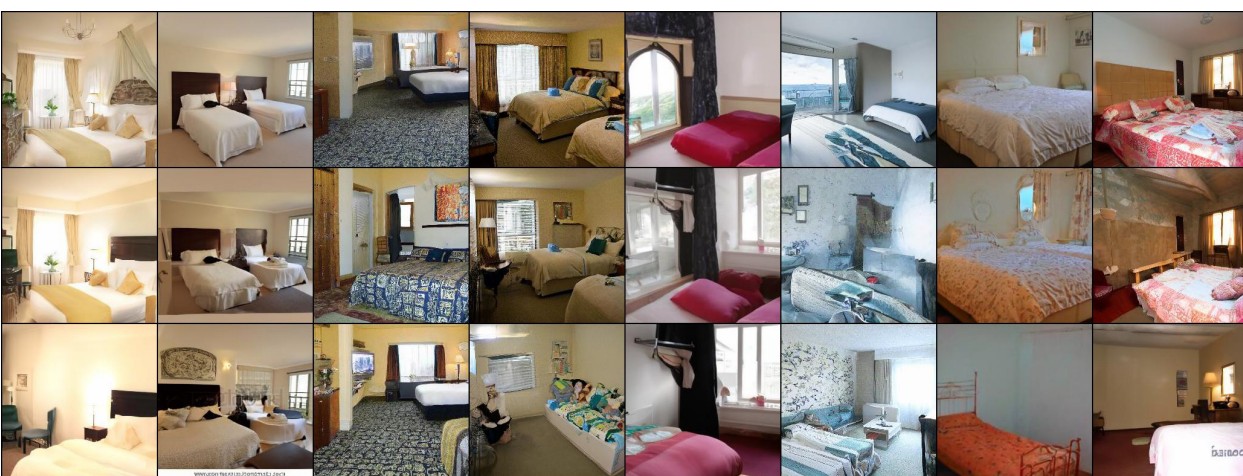

Figure 14: **Images generated by ADM pre-trained on LSUN-Bedroom with per-sample perturbation.** (**Middle**): Images generated by the pre-trained model. (**Top**): Generated by the same model with perturbed latent vectors for better realism. (**Bottom**): Generated by the same model with perturbed latent vectors for better diversity.

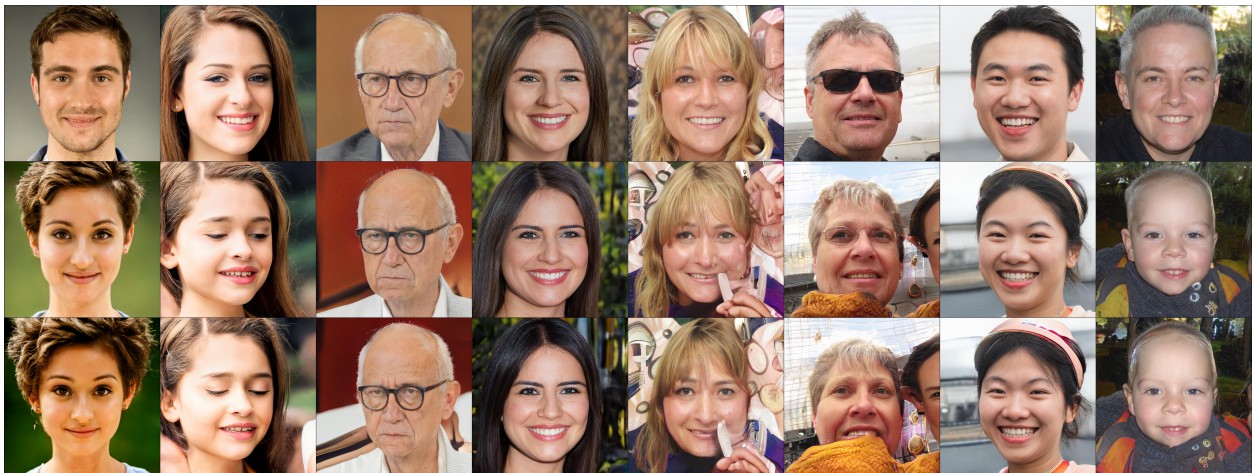

Figure 15: **Images generated by StyleGAN2 trained on FFHQ after fine-tuning with importance sampling.** Within each column, images are generated with the same latent code. (**Middle**): Images generated by the pre-trained model. (**Top**): Generated by the model fine-tuned for better fidelity. (**Bottom**): Generated by the model fine-tuned for better diversity.

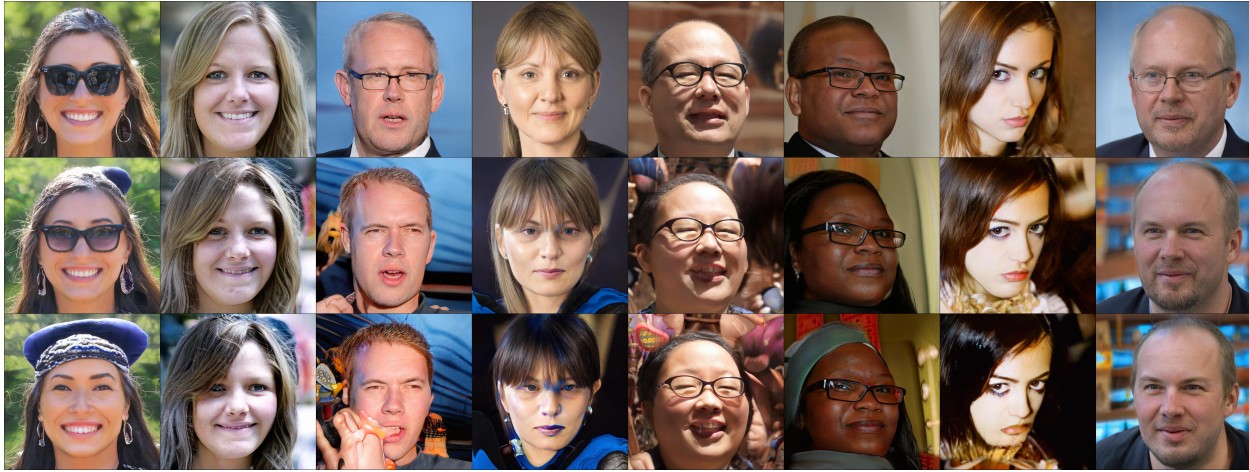

Figure 16: **Images generated by StyleGAN3-T trained on FFHQ after fine-tuning with importance sampling.** Within each column, images are generated with the same latent code. (**Middle**): Images generated by the pre-trained model. (**Top**): Generated by the model fine-tuned for better fidelity. (**Bottom**): Generated by the model fine-tuned for better diversity.

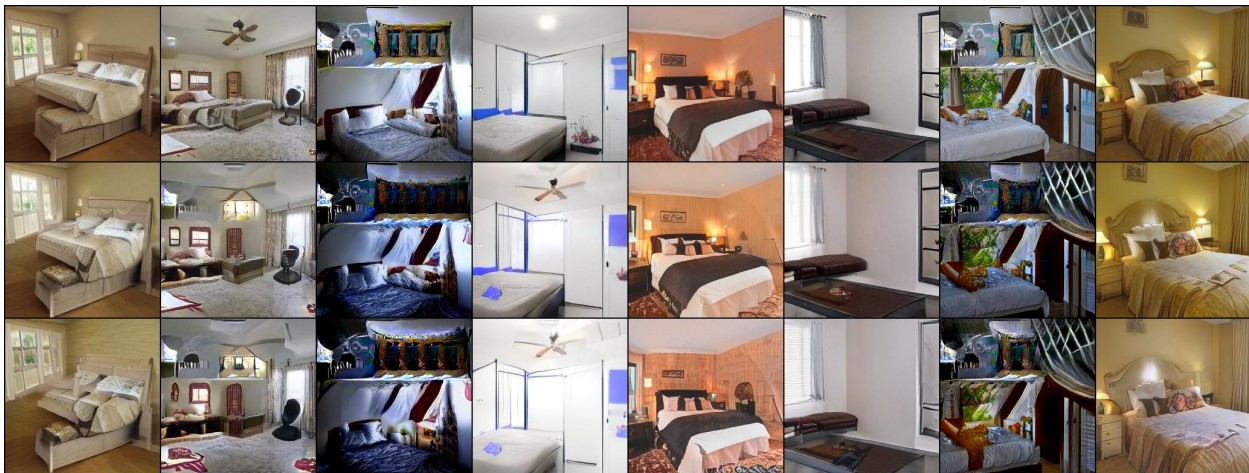

Figure 17: **Images generated by ProjectedGAN trained on LSUN-Bedroom after fine-tuning with importance sampling.** Within each column, images are generated with the same latent code. (**Middle**): Images generated by the pre-trained model. (**Top**): Generated by the model fine-tuned for better fidelity. (**Bottom**): Generated by the model fine-tuned for better diversity.

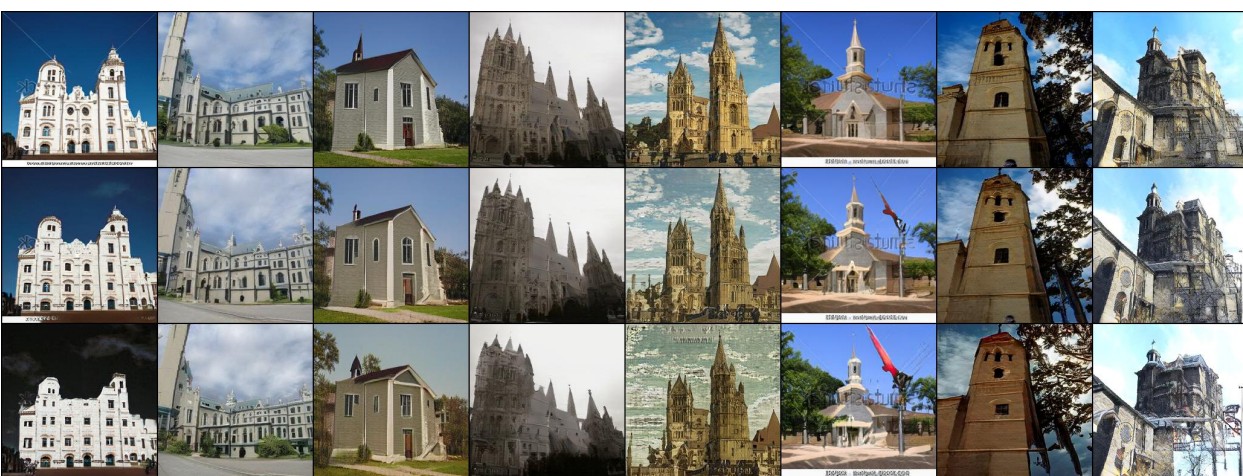

Figure 18: **Images generated by ProjectedGAN trained on LSUN-Church after fine-tuning with importance sampling.** Within each column, images are generated with the same latent code. (**Middle**): Images generated by the pre-trained model. (**Top**): Generated by the model fine-tuned for better fidelity. (**Bottom**): Generated by the model fine-tuned for better diversity.

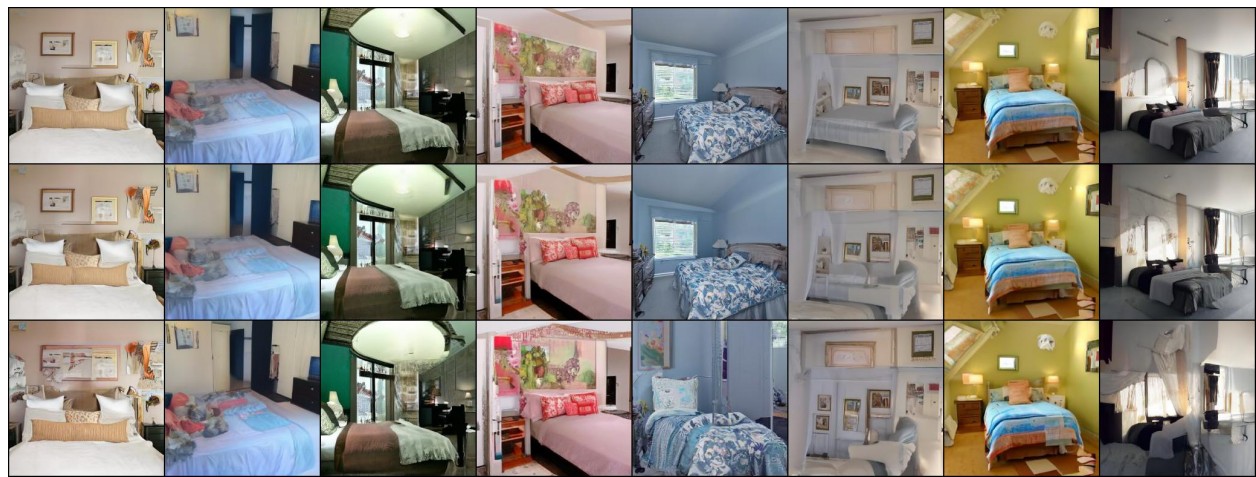

Figure 19: **Images generated by IDDPM trained on LSUN-Bedroom after fine-tuning with importance sampling.** Within each column, images are generated with the same latent code. (**Middle**): Images generated by the pre-trained model. (**Top**): Generated by the model fine-tuned for better fidelity. (**Bottom**): Generated by the model fine-tuned for better diversity.

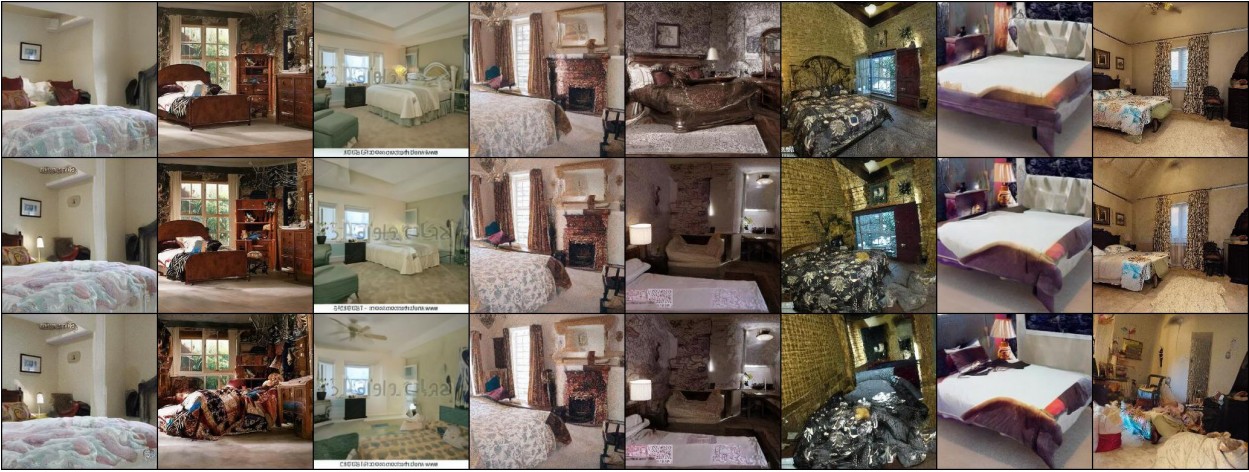

Figure 20: **Images generated by ADM trained on LSUN-Bedroom after fine-tuning with importance sampling.** Within each column, images are generated with the same latent code. (**Middle**): Images generated by the pre-trained model. (**Top**): Generated by the model fine-tuned for better fidelity. (**Bottom**): Generated by the model fine-tuned for better diversity.

