# OpenReview forum: "Controlling the Fidelity and Diversity of Deep Generative Models via Pseudo Density"
_TMLR — Accepted by TMLR_

### Review · Reviewer_vAPv · 2024-04-11

**Summary Of Contributions:**

This paper introduces a technique to leverage importance sampling to increase fidelity or diversity in newly generated samples from generative models. Crucially, the authors introduce the concept of pseudo-density, which they present as a way to attain a probability density for the target distribution. With this tool at hand, importance sampling, e.g., reweighting/rejection sampling, can be applied both at inference and training (fine-tuning) time to manipulate the model's output to achieve a more diverse or realistic sampling of new images.

While I refrain from judging the novel of this work as my knowledge of related literature/work in the field of computer vision is somehow limited, I do find this work to be, in general, very interesting and its contribution to be of valuable interest to the machine learning community.

**Audience:**

Yes

**Broader Impact Concerns:**

I couldn't identify any specific ethical concerns related to the submitted work that need explicit mention. However, all general ethical implications associated with training and utilizing a generative model for image generation, such as potential bias embedded in the model, copyright issues, and privacy concerns within the training data, still hold true.

**Claims And Evidence:**

Yes

**Requested Changes:**

- It would be important to have some cartoons that sketch the basic idea behind the pseudo density and how it is computed. This would help the reader to have an intuitive understanding of this new metric. In this regard, I find the first paragraph of section 3 a bit hard to parse. The clarity of this paragraph can be easily enhanced by elaborating a bit more on the details and by providing a visual example.

- At the bottom of page 3, the authors introduce the volume a sample occupies $V_i^k$, which is proportional to a constant $C$. This constant is said to depend on the dimensionality of the feature space $n$, although this dependence is not made explicit in the formulae. This might be a little oversight unless I misunderstood something. It'd be great if the author could edit the formulae making the dependence of $C$ on $n$ clear. On the other hand, I find this a bit counterintuitive because having a large feature space should increase the capability of the pseudo-density network to be more expressive. Or is my intuition off here?

- In Fig. 1, the caption indicates that the four left-most images have the highest pseudodensity, while the four right-most images have the lowest. I suggest making this distinction clearer in the figure itself. As a recommendation, the authors could separate the two blocks of 3x4 images, inserting dots in the middle to indicate that there are images lying in between. Additionally, they could add an "x-axis" at the bottom displaying pseudodensity as an increasing quantity from right to left (or rearrange the blocks and depict the arrow moving from left to right for increasing pseudodensity, which I find more natural). The authors can refer to Fig. 2 as an example of what I mean.

-  Small typo: $F(x_1 \to F(x_1)$ (page 4)
- Small typo: where $\epsilon$ is the predefined the adversarial budget $\to$ remove **the** (page 5)

- The first sentence of section 5.2 is not clear to me. It'd be great if the authors could rephrase it.

- In the last sentence on page 6, the authors state that GANs not only have access to the real data distribution but also to the **generated data** distribution. What do the authors mean here? Neither GANs nor diffusion models (in general) give you access to the probability density (or likelihood) for the generative distribution. In this respect, I don't see any difference between those two approaches. Am I misunderstanding something?


- The top paragraph on page 7 is not very clear to me. First, I am confused by $Pert_g(\cdot)$ in the third line. Should not this be $Pert_r(\cdot)$? Second, I do not fully understand why we need to perform importance sampling of the real data distribution. Importance sampling is used to  *reweight* a probability density so that it matches closely to a target density. In that sense, I do expect to use IS only to *reweight* generated samples based on the pseudodensity as introduced in the paper. I would appreciate the authors elaborating on this and clarifying this paragraph. Again, some more detailed formulae and/or a cartoon would help clarify this crucial part of the paper as well.

- The last paragraph of section 5 seems to suggest that high fidelity and high diversity can be achieved when the importance sampling strategy and other fine-tuning strategies, such as truncation, are applied together. However, not many details are provided in the main manuscript. The authors defer to Appendix A.4, but I did not find much information there. It'd be great if the authors could expand on this, elaborating on the limitations of this method. If there are no limitations in principle I wonder why the authors have not dedicated a greater focus on this combined approach in their experiments.

- small typo: manipulate $\to$ manipulates (page 8)

- In Figure 3, what do the authors mean with *Ours Frontier* in the plot legend?

## More conceptual questions

- Does the network computing the pseudo-density from the extracted features need to be trained independently for each dataset considered, or is it rather a one-fits-all kind of network that can be trained once (e.g., on Imagenet) and applied to different datasets?

- In page 4, below the figure, the author say that $n$ is set small, e.g., 1,2. Is there any practical heuristics to choose the dimensionality of the feature space? What's the effect of setting $n$ to larger values? Does the quality deteriorate substantially? I presume the authors want to keep the Volume of the ipersphere $V^k_i$ to be small as this is inversely proportional to the pseudodensity. Is my intuition correct?

- As a naive question to the authors: indeed, manipulating the density to produce more diverse samples is likely increasing the number of defects and weird artifacts in the image. Does the author have in mind any technique that could help achieve a high diversity while limiting this artifact, which deteriorates the quality of the generated sample?

- Throughout the paper, the authors discuss different types of generative models such as GANs and DDPMs. However, they never mention autoregressive models, such as PixelCNN. While those have their limitations and may not perform as well as SOTA GANs and DDPMs, they give access to the exact likelihood of the generative distribution. Therefore, combined with the pseudodensity applied to the real data, one can efficiently compute the importance weights and, therefore, perform the same type of rejection sampling and/or fine-tuning, even in a more natural way (since the generative likelihood is given). Is there any specific reason why the authors have not considered such kind of generative models?

**Strengths And Weaknesses:**

**Strenghts**

- The paper reads well and is scientifically sound.
- The results presented in this work are interesting, and the approach of using importance sampling seems quite effective in the analysis performed in the experiment section.

**Weaknesses**
- I think the paper requires an intuitive visualization of the pseudo-density to enhance clarity and make it easier for the general reader to grasp the basic idea immediately.
I don't have further weaknesses to highlight. The paper appears to be solid work.
- However, I do have changes to request, as well as some more conceptual questions to address the author. Please see the section below **Requested Changes**.

---

> ### Author Response · Authors · 2024-07-23
>
> We sincerely thank the reviewer for their thorough feedback and insightful questions, which have significantly improved our paper. We have addressed the points raised as follows:
>
> - Requested Changes:
>
> > **Visualize the basic idea and clarify the first pargraph of section 3.**
>
> A: We have added a new figure (Figure 1 in the updated PDF) to help readers understand the intuition of pseudo density and how it is computed.
>
> > **Making the dependence of C on n clear. Should having a large feature space (n)  increase the capability of the pseudo-density network?**
>
> A: We thank the reviewer for pointing out the oversight regarding the volume expression. We have added an explicit expression of the volume to clarify this point. We also clarify that the capability of the pseudo-density network is independent of the value of n. This is because n is only relevant for computing the pseudo-density value for each sample, which serves as the target for training the network.
>
> > **Improve Figure 1 (now Figure 2) for clarity.**
>
> A: We have updated this figure as suggested to clarify the distinction between high and low pseudo density images.
>
> > **Typos in page 4 and 5.**
>
> A: We have fixed the typos as pointed out.
>
> > **Unclear first sentence of section 5.2.**
>
> A: We have rephrased the first sentence of section 5.2 to: "Using sampling at inference to modify the generated samples introduces additional computational overhead and reduces model throughput. An alternative approach is to fine-tune the model, which achieves the same effect (i.e., enhancing its diversity or fidelity) without increasing inference time."
>
> > **Clarify the last sentence on page 6 on the generated data distribution. Why only GAN and not diffusion?**
>
>
> A: We clarify that our method modifies the distribution of training samples in generative models to bias their generated outcomes. The training samples for a diffusion model or the generator of a GAN model are **real** samples, which we apply important sampling on. Additionally, when training the discriminator of a GAN model, we also consider modifying the distribution of the **generated** samples. However, it is exclusive to GAN models and not relevant to diffusion models, as they do not involve a discriminator. We have clarified this in the updated manuscript.
>
> > **Confusion about notation (Should Pert_g be Pert_r?) and the need for importance sampling of real data distribution. I expect to use IS only to reweight generated samples based on the pseudodensity as introduced in the paper.**
>
>
>
> A: We clarify that the final goal is to enhance the diversity or fidelity of the generated samples. We achieve this by adaptively sampling the training data, including both real and generated samples. For example, sampling more real samples with low density scores would bias the model to generate those types of samples more frequently, and vice versa. Directly forcing the distribution of generated samples to follow a certain distribution through our importance sampling is feasible, similar to our approach in section 5.1 where we perform inference with importance sampling. However, in the context of section 5.2, we describe a fine-tuning approach that indirectly modifies the distribution of the generated samples through training data modification.
>
> > **It seems to have no limitation in principle when combining the importance sampling strategy and other fine-tuning strategies, such as truncation. Should we investigate more?**
>
> A: We clarify that it is not possible to simultaneously achieve both high diversity and high fidelity with current methods. All methods, includeing ours, involve a trade-off between fidelity and diversity.  When using our method in conjuntion with existing techniques such as the truncation trick, we obtain a more effective and flexible system to control the generated sample distribution, as demonstrated in Appendix A.4.
>
> > **Typo: "manipulate" should be "manipulates".**
>
> A: We have fixed the typo "manipulate" to "manipulates".
>
> > **Confusion about "Ours Frontier" in Figure 3 (now Figure 4)**
>
> A: In Figure 3 (now Figure 4), "Frontier" refers to the Pareto frontier, the optimal trade-offs achieved across various hyperparameter configurations. We have clarified this in the figure caption.

---

> ### Author Response · Authors · 2024-07-23
>
> - Conceptual Questions:
>
> > **Does the pseudo-density network need to be trained independently for each dataset?**
>
> A: Yes, the network computing the pseudo-density needs to be trained independently for each dataset considered. We have not explored a one-fits-all approach, though theoretically it might work if trained on a sufficiently diverse dataset.
>
> > **How to choose n? What's the effect of larger values?**
>
> A: We have found that setting n=1 or 2 works well and yields very similar results. Setting n too large leads to numerical instability (very small and very large density values) and substantially deteriorated performance due to the difficulty in fitting the density network.
>
> > **Any technique to achieve high diversity while limiting artifacts?**
>
> A: We are not aware of any technique that can help achieve high diversity while limiting artifacts, except for further training the model on addithigher-quality data.
>
> > **Why weren't autoregressive models considered?**
>
> A: We thank the reviewer for suggesting the exploration of autoregressive models. In response, we have added a new section, Appendix A.5, where we compare pseudo-density-based sampling with log-likelihood-based sampling using the autoregressive VQGAN[1]. Our results show that log-likelihood-based sampling yields a worse precision-recall tradeoff than pseudo-density-based sampling and fails to enhance the diversity of generated samples. This indicates that images with higher likelihood tend to have better realism, but ones with lower likelihood do not have better uniqueness. We conclude that the likelihood metric provided by autoregressive models is less effective than pseudo density for controlling fidelity and diversity in generated samples.
>
> We appreciate the reviewer's valuable input, which has helped us strengthen our work and expand its scope. We hope these changes and clarifications address all the raised concerns and improve the paper's overall quality and clarity.
>
> [1] Esser, Patrick, Robin Rombach, and Bjorn Ommer. "Taming transformers for high-resolution image synthesis." *Proceedings of the IEEE/CVF conference on computer vision and pattern recognition*. 2021.

---

### Review · Reviewer_fPN1 · 2024-05-31

**Summary Of Contributions:**

The authors propose several methods to control the fidelity-diversity tradeoff of a wide class of unconditional generative models. The core of all the proposed methods is transforming real data to the feature space, empirically estimating density in this feature space based on the nearest neighbors approach, and then approximating this density by a neural network. Thus, the authors propose to learn a model that assigns a so-called pseudo-density for any given data point. Then the authors propose 3 specific methods to control fidelity-diversity tradeoffs based on this pseudo-density model. The first one is applicable to the learned model and implies optimizing latent code z to move it to the regions of high pseudo-density to improve fidelity or to the regions of low pseudo-density to improve diversity. The second and third methods combine rejection and importance sampling using pseudo-density. As well as in the previous case, using objects from the region of high density leads to better fidelity, while using objects from the region of lower density leads to better diversity. The second and third methods are similar, but the second method is used at the inference stage with the learned model, while the third method is used for fine-tuning.

**Audience:**

Yes

**Claims And Evidence:**

Yes

**Requested Changes:**

I am not deeply aware of related works in the subfield of controlling the fidelity-diversity trade-off in generative models so I can not truly judge how comprehensive the provided comparison is. Anyway, considering the other mentioned methods [1, 2] and studying the computational cost is highly appreciable since the proposed approach is mostly empirical. (This would strengthen the work in my view).

[1] Tanaka A. Discriminator optimal transport //Advances in Neural Information Processing Systems. – 2019. – Т. 32.

[2] Azadi S. et al. Discriminator rejection sampling //arXiv preprint arXiv:1810.06758. – 2018.

**Strengths And Weaknesses:**

**Strengths:**
- The paper is well-written and provides a clear description of the proposed methods and obtained results.
- The proposed methods are simple and can be directly applied to a learned generative model from a wide class of models.
- The authors provide examples of their approach applied to different generative models, including several GANs and one diffusion model.
- The authors demonstrate that their approach is, in some cases, superior to other baseline methods in terms of the fidelity-diversity Pareto frontier.

**Weaknesses:**
- The comparison is limited. The authors compare their methods only with the truncation method and polar sampling. Furthermore, polar sampling is considered only with a fraction of the models used for comparison.
- Some of the competitive methods [1,2] are not considered. The authors state that such methods have relatively higher computational costs than their method. However, they do not support this claim by studying computational costs.

[1] Tanaka A. Discriminator optimal transport //Advances in Neural Information Processing Systems. – 2019. – Т. 32.

[2] Azadi S. et al. Discriminator rejection sampling //arXiv preprint arXiv:1810.06758. – 2018.

---

> ### Author Response · Authors · 2024-07-22
>
> We appreciate the reviewer's thoughtful comments and suggestions. We have made changes in the updated PDF and would like to address the point raised in the Weaknesses section.
>
> > **Q1: Polar sampling is considered only with a fraction of the models used for comparison.**
>
> A: This is due to the extended preprocessing times required by this method. For example, using polarity sampling with a StyleGAN2 model necessitates 14 days for preprocessing. Therefore, we cannot conduct this experiment for all the models.
>
> > **Q2.1: Some of the competitive methods [1,2] are not considered.**
>
> A: We thank the reviewer for the suggestion. We have added in our updated PDF a comparison of our approach with the methods Discriminator Optimal Transport [1] and Discriminator Rejection Sampling [2] in Appendix A.2. The results indicate that methods relying on the trained discriminator for enhancing image realism do not perform well for high-resolution image data and larger, modern GAN architectures like StyleGAN.
>
> > **Q2.2: Computational costs.**
>
> A: We have added a detailed analysis of the extra computational cost in Appendix A.3. This analysis compares our approach with [1][2], and polar sampling, supporting our claim about the relative efficiency of our method. Our inference-time method outperforms the compared methods in terms of either sampling efficiency or preprocessing time. Our fine-tuning method does not decrease sampling efficiency or require preprocessing for sampling.
>
>
> [1] Tanaka A. Discriminator optimal transport - NeurISP 2019
>
> [2] Azadi S. et al. Discriminator rejection sampling. arXiv preprint arXiv:1810.06758. – 2018.

---

### Review · Reviewer_gu77 · 2024-07-11

**Summary Of Contributions:**

The paper proposes a method of modifying the distribution of samples produced by a generative model with the goal of varying the realism of generated images or the overall diversity in the overall distribution. This is achieved by proposing a metric called pseudo density that aims to approximate the underlying probability distribution of the dataset samples.

To do so, probability densities are first approximated at individual images of the training dataset using a k-nearest neighbor based scheme. Then, a fully connected network is fitted over them to infer the densities at arbitrary images. This approximation obtained using the fully connected network at any data point $x$ is termed pseudo density.

This proposed metric can be used to modify the individual generated samples and make them more realistic by locally perturbing them with the goal of increasing their pseudo density. Additionally, the generated distribution can be altered during inference by assigning different weights to images depending on their pseudo density followed by using an importance sampling based strategy. A similar approach is proposed to fine-tune a trained model to produce samples with greater realism vs greater diversity.

The efficacy of the proposed approach is validated experimentally using different generative models.

**Audience:**

Yes

**Claims And Evidence:**

Yes

**Requested Changes:**

Please see the section above

**Strengths And Weaknesses:**

**Strengths**

-	The proposed framework is a simple and intuitive way to bias generative models towards higher fidelity or diversity.
-	The experimental results are largely positive and, show interpretability in the controls provided by pseudo density.
-	As someone who is not very familiar with this area, I enjoyed reading this paper. It is generally well written and easy to follow.

**Weaknesses**

I do not have any major weakness to point out but would like to clarify something about the result in Figure 4.

Unlike in other cases, some of the face images appear to provide less realistic outputs on fine tuning with a higher w, when one would expect otherwise. Did the authors explore why that is in their experiments?

I wonder if the nearest neighbors of these images in the training dataset (after feature embedding) are more further apart than usual, thereby resulting in poor approximation of pseudo density at these test images.

---

> ### Author Response · Authors · 2024-07-22
>
> We appreciate the reviewer's thorough reading of our paper and their positive comments on the framework's simplicity, intuitive nature, and the overall experimental results. We would like to address the point raised in the Weaknesses section regarding Figure 4 (now Figure 5 in the updated PDF).
>
> > **Q1: "Unlike in other cases, some of the face images appear to provide less realistic outputs on fine tuning with a higher w, when one would expect otherwise."**
>
> A: We believe that the reviewer is referring to the overly simplistic backgrounds in those cases, which might make them appear less realistic. Indeed, we observe that with a higher value of w, the model tends to omit various details, such as complex backgrounds or accessories (e.g., glasses or headbands). Nonetheless, it is important to note that this also reduces the likelihood of artifact generation, thereby enhancing overall realism. Specifically, for the four left-to-right cases illustrated in Fig. 4, the artifacts exhibited in the orginal images that were removed when fine-tuning with a high value of w are:
>
> -- 1st image: an artefact akins to a headband.
>
> -- 2nd image: noisy background and body part
>
> -- 3rd image: an artefact at the ear
>
> -- 4th image: additional hair in the background
>
> Furthermore, as can be seen from the figure, when fine-tuning with a lower value of w, the model tends to add more details, which increases overall diversity but also at the cost of artifact generation. This trend is consistent across all use cases.
>
> > **Q2: "I wonder if the nearest neighbors of these images in the training dataset (after feature embedding) are more further apart than usual, thereby resulting in poor approximation of pseudo density at these test images."**
>
> A: No, we didn't observe that phenomenon. In fact, we note that the images with simpler backgrounds tend to obtain higher pseudo density values, thus are surrounded by nearer neighbors than usual in the feature space. Conversely, those images with more artefacts tend to have less near neighbors. We didn't observe degraded approximation of pseudo density for samples that have fewer near neighbors.

---

### Decision · Action_Editor_DVCV · 2024-09-10

**Recommendation:** Accept with minor revision

**Comment:**

The paper provides a sound and intuitive approach to controlling fidelity and diversity in generative models. The introduction of pseudo density as a control mechanism is novel in its application and shows clear empirical benefits across a range of models. In particular, the results are well-supported by extensive experiments across multiple generative models, including GANs and diffusion models. The approach demonstrates flexibility and control, and the empirical evidence for the claims is largely convincing. While the paper is more empirical than theoretical, the experiments validate the effectiveness of the method.

The authors have effectively addressed most of the reviewers' concerns in the rebuttal. The authors are encouraged to incorporate all the requested changes from reviewers as well as their clarifications in the response. For example, further clarity in explanation of certain sections, especially around the pseudo-density concept and the impact of feature space dimensionality would be great. Some visual improvements could be made to make the figures more intuitive. Overall, the paper makes a valuable contribution with well-supported claims. Experimental validation is strong, and most of the reviewers' concerns have been addressed. A final round of polish could enhance the clarity and presentation of the paper.

**Audience:**

This work would be interesting to the entire generative model community

**Claims And Evidence:**

This paper proposes methods to manipulate the fidelity-diversity trade-off in generative models like GANs and diffusion models. The key contribution is introducing a metric called pseudo density, which approximates the likelihood of individual samples using nearest-neighbor data from real samples. The authors present three techniques to control fidelity and diversity: 1) Per-sample perturbation to adjust individual sample features for fidelity or diversity. 2) Importance sampling during inference to bias the generated data towards higher fidelity or diversity. 3) Fine-tuning with importance sampling to alter the generative model’s distribution and improve performance metrics like the Frechet Inception Distance (FID). The paper demonstrates improvements in model performance and control over fidelity-diversity trade-offs, showing the flexibility of their approach across various generative models, including several GAN and diffusion models.